# Sex disparate gut microbiome and metabolome perturbations precede disease progression in a mouse model of Rett syndrome

Kari Neier[1], Tianna E. Grant[1], Rebecca L. Palmer[1], Demario Chappell[1], Sophia M. Hakam[1], Kendra M. Yasui[2], Matt Rolston[1], Matthew L. Settles[3], Samuel S. Hunter[3], Abdullah Madany[1], Paul Ashwood[1], Blythe Durbin-Johnson[3,4], Janine M. LaSalle [1,3,5 ✉] & Dag H. Yasui [1,5]

Rett syndrome (RTT) is a regressive neurodevelopmental disorder in girls, characterized by multisystem complications including gut dysbiosis and altered metabolism. While RTT is known to be caused by mutations in the X-linked gene *MECP2*, the intermediate molecular pathways of progressive disease phenotypes are unknown. *Mecp2* deficient rodents used to model RTT pathophysiology in most prior studies have been male. Thus, we utilized a patient-relevant mouse model of RTT to longitudinally profile the gut microbiome and metabolome across disease progression in both sexes. Fecal metabolites were altered in *Mecp2e1* mutant females before onset of neuromotor phenotypes and correlated with lipid deficiencies in brain, results not observed in males. Females also displayed altered gut microbial communities and an inflammatory profile that were more consistent with RTT patients than males. These findings identify new molecular pathways of RTT disease progression and demonstrate the relevance of further study in female *Mecp2* animal models.

[1] UC Davis School of Medicine, Department of Medical Microbiology and Immunology, Genome Center, MIND Institute, Davis, CA, USA. [2] Oregon State University, Corvallis, OR, USA. [3] UC Davis Genome Center, Davis, CA, USA. [4] UC Davis School of Medicine, Department of Public Health Sciences, Davis, CA, USA. [5] These authors contributed equally: Janine M. LaSalle, Dag H. Yasui. ✉email: jmlasalle@ucdavis.edu

Rett syndrome (RTT, OMIM #312750) is one of the most common genetic causes of intellectual disabilities in females, affecting 1 in 10,000 births[1]. RTT is an X-linked dominant disorder that is predominantly caused by mutations in *MECP2*[2], a gene encoding Methyl-CpG Binding Protein 2 (MeCP2). Although RTT is a monogenic disorder, its molecular pathogenesis remains poorly understood, particularly in *Mecp2* heterozygous females. Further complexities of RTT are likely due to pleiotropic molecular functions and ubiquitous expression of MeCP2. MeCP2 was originally characterized as a methyl CpG binding protein and transcriptional repressor[3]. More recently, MeCP2 has been shown to; (1) bind to additional nucleotide motifs across the genome[4–8], (2) activate as well as repress gene transcription[9,10] including microRNAs (miRNA)[11,12], (3) regulate alternative splicing[13], and (4) nucleate higher-order chromatin organization[14]. Furthermore, X-chromosome inactivation results in mosaic expression of *MECP2* mutations in females, and cells expressing mutant *MECP2* have non-cell autonomous, or "bad neighborhood" effects, negatively impacting nearby wild-type cells in Rett females[15–17]. RTT patients are predominantly heterozygous females because spontaneous *de novo* mutations occur more frequently on the paternal X-chromosome[18] and early lethality occurs in boys with RTT-causing *MECP2* mutations.

Despite the vast majority of RTT patients being female, the majority of gene therapy and other preclinical studies in animal models of RTT have been conducted in male mice hemizygous for *Mecp2* deletion. The complexities of RTT and MeCP2 function have resulted in significant challenges for developing safe and effective gene therapies[19], and it is unclear whether gene therapies given to existing RTT patients would effectively mitigate systemic manifestations of the disease. Thus, well characterized, gene-relevant female *Mecp2* mosaic mouse models are needed to uncover underlying molecular, cellular, and physiological intermediate phenotypes in the pathophysiology of RTT in order to provide insights into potential therapies.

Individuals with RTT initially present an apparently normal phenotype, with distinct symptoms emerging later at 6-18 months of age that include regression of motor and language skills, seizures, microcephaly, loss of purposeful hand movements, breathing abnormalities and severe cognitive impairments[20,21]. The neurologic features of RTT in conjunction with the importance of MeCP2/*MECP2* in neuronal development and function in adulthood have prompted intense research focus on the central nervous system in RTT patients and mouse models. However, RTT has been increasingly recognized as a multi-system disorder likely due to the expression of *MECP2* in almost all cell types. For example, approximately two-thirds to four-fifths of RTT patients frequently report gastrointestinal disturbances[22,23], and half of RTT patients have high serum cholesterol, triglycerides, and/or LDL cholesterol[24]. Male, *Mecp2*$^{-/y}$ hemizygous mice with deletion of *Mecp2* exons 3 and 4 (*Mecp2*-null) showed severe defects in the colonic epithelium[25], and an N-ethyl-N-nitrosourea mutant suppressor screen revealed a significant role for *Mecp2* in lipid metabolism[26].

The gut microbiome has important roles in physiological function of the GI tract, energy metabolism, the immune system, and can modulate brain and behavior[27]. Furthermore, changes in bacterial genus abundance correlate with neurologic phenotypes in schizophrenia[28], major depressive disorder[29] and Parkinson's disease[30]. Extensive cross-talk between the gut and the nervous system involving gut microbes and the metabolites they produce[31–33] suggests a potential role for the gut microbiome in RTT. Recent clinical studies have found that the gut microbiome is perturbed in RTT patients[34,35]. However, whether the altered microbiota in RTT patients is simply a consequence of their altered diet and feeding behavior or an important contributor to regression is currently unknown. Thus, examination of the gut microbiome and metabolome in a female mouse model of RTT throughout the course of disease progression would provide new critical insights into RTT pathophysiology.

RTT is a progressive neurologic disorder with age-specific manifestations. RTT is divided into four stages: early stagnation, rapid regression, stabilization, and late motor deterioration[36]. In addition, motor skill defects, mood, and gastrointestinal problems vary with increased age in RTT patients[23,37]. Mouse *Mecp2* deficient models of RTT recapitulate progression of motor, behavioral, and metabolic phenotypes. Both male *Mecp2*$^{-/y}$-null and female *Mecp2*$^{-/+}$ heterozygous mutant mice exhibit age- and sex-specific increases in body weight, gait alterations, reduced anxiety behaviors, and decreased performance on beam walking and rotarod tasks[38–40]. Time-dependent signatures of RTT can also be detected at the neuronal level. For example, GABA signaling is decreased in an age-specific manner in *Mecp2*$^{-/y}$ null males[41], and dendritic spine density in *Mecp2*$^{-/y}$ relative to *Mecp2*$^{+/y}$ control males is progressively decreased with age reflecting time-dependent behavioral phenotypes[42]. Furthermore, neuronal MeCP2 expression levels increase with age[17,43]. While there is substantial evidence of biochemical and biomolecular time-dependent signatures of RTT in the brain, molecular signatures of gastrointestinal and metabolic disease progression have not previously been thoroughly examined.

We sought to characterize gastrointestinal and metabolic molecular signatures of RTT disease progression at the gut-brain interface by simultaneously evaluating the fecal microbiome and metabolome longitudinally across symptom progression in a construct-relevant mouse model of RTT, that leverages an *Mecp2-e1*-isoform-specific knock-in mutation found in human RTT patients[38,40]. We examined both male hemizygous *Mecp2-e1*$^{-/y}$ mice and female mosaic heterozygous *Mecp2-e1*$^{-/+}$ mice to identify sex-specific differences in RTT disease progression. Fecal collection allowed us to longitudinally analyze microbiome and metabolome profiles from the same mice across disease progression, and we utilized weighted network construction analyses to integrate molecular and phenotypic data. Lastly, we measured the brain lipidome in male and female mutant *Mecp2-e1* mice to determine whether changes in the gut microbiome and metabolome were associated with altered lipid profiles in the brain. These findings demonstrate that pathophysiology and progression of RTT is substantially different in females than in males and suggest that changes in the microbiome and metabolism of the gastrointestinal tract influence progression of overt neurological, motor, and metabolic symptoms in RTT, providing new avenues for potential treatments and therapies.

## Results

**Mice deficient in MeCP2-e1 display sexually dimorphic progression of neurological, motor, and body weight phenotypes.** To longitudinally measure RTT disease progression in *Mecp2-e1*$^{-/+}$ and *Mecp2-e1*$^{-/y}$ mice, weekly neurological, motor, and metabolic phenotyping assessments were carried out on mutant and wild-type littermates between 5–6 and 19 weeks of age for females and 5–6 and 16 weeks of age for males. Male *Mecp2-e1*$^{-/y}$ mice began to experience increased morbidity and mortality at approximately 16 weeks of age, so measurements ceased at 16 weeks and mice were euthanized as required by IACUC protocol. Neurological phenotyping scores were assigned based on a system previously used to assess disease progression in RTT mouse models[40,44], motor phenotyping of gait was performed by footprint analysis, and gross metabolic phenotype was assessed by body weight measurements.

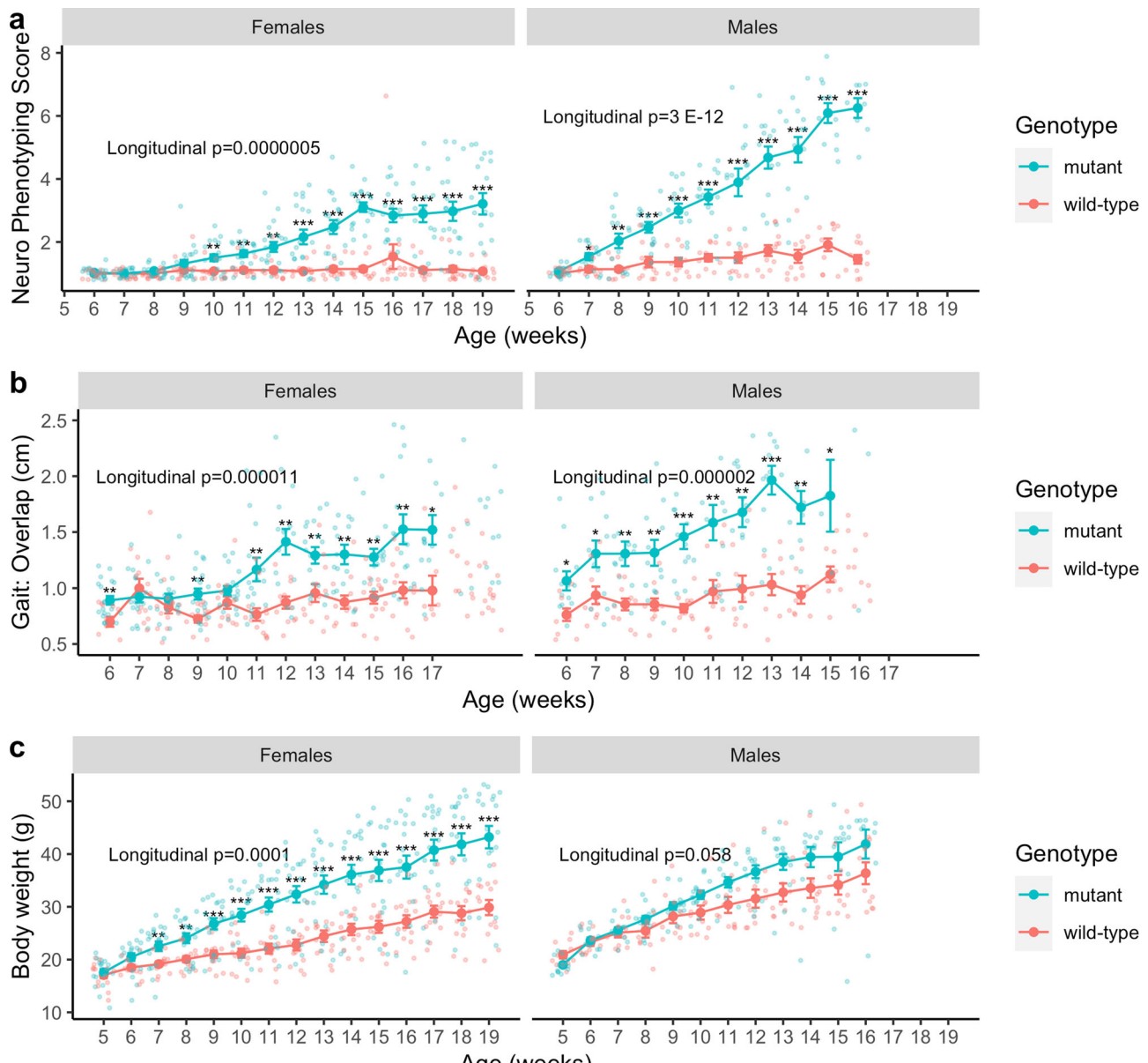

**Fig. 1 Longitudinal phenotypes in *Mecp2-e1* mutant and wild-type mice. a** Comparison of neurophenotyping score (range 1–7, 1 being least severe and 7 being most severe) in *Mecp2-e1* mutant vs. wild-type (wt) mice longitudinally and cross-sectionally across disease course (between 6 and 19 weeks of age for females and 6 and 16 weeks of age for males), stratified by sex and controlling for within-litter effects. **b** Comparison of the most sensitive measure of gait (overlap distance in centimeters between hind and front legs, with a greater distance indicating a more impaired gait) in mutant vs. wild-type mice longitudinally and cross-sectionally between 6 and 17 weeks of age in females and 6 and 15 weeks of age for males, stratified by sex and controlling for within-litter effects. **c** Comparison of body weight of mutant vs. wild-type mice longitudinally and cross-sectionally between 5 and 19 weeks of age for females and 5 and 19 weeks of age for males, stratified by sex and controlling for within-litter effects. Cross-sectional analyses were controlled for multiple comparisons at each time point. N = 11–19/genotype/sex. Each solid dot represents the mean of each group at each time point and error bars are standard error of the mean (SEM). *FDR < 0.05, **FDR < 0.01, ***FDR < 0.001 in mutant vs. control, cross-sectionally. Longitudinal *p*-values represent the overall association between mutant and wild-type mice across disease course using linear mixed effects models.

As expected both female *Mecp2-e1⁻/⁺* and male *Mecp2-e1⁻/y* mice exhibited progression of neurological phenotypes compared to wild-type littermates (longitudinal p < 0.0001, both sexes), however onset was earlier and disease severity was greater for males than for females (Fig. 1a), as previously described[38]. The earliest age at which there was a statistically significant increase in neuro-phenotyping scores was at 10 weeks for females and at 7 weeks for males, with males displaying greater severity of neurological symptoms, as expected.

In congruence with neurological motor phenotyping outcomes, gait analyses performed from 6 to 17 weeks in females and 6 and 15 weeks in males indicate that while both female and male *Mecp2-e1* mutant mice exhibited progressive motor phenotypes compared to *Mecp2* wild-type control littermates, symptom onset was earlier, and severity was greater for males (Fig. 1b). In females, *Mecp2-e1⁻/⁺* mutants had larger foot overlap distances between the front and hind legs than *Mecp2⁺/⁺* littermates at 6 and 9 weeks, with persistent differences beginning at 11 weeks of

**Table 1 Longitudinal analyses of relative phyla abundance and diversity scores by *Mecp2-e1* Genotype.**

|  | Females | | Males | |
| --- | --- | --- | --- | --- |
| Phylum | Beta (%) | p-value | Beta (%) | p-value |
| **Firmicutes** | **3.72000** | **0.033** | **−5.13000** | **0.010** |
| **Bacteroidetes** | **−3.36000** | **0.038** | **5.23000** | **0.0085** |
| **Actinobacteria** | −0.02950 | 0. 26 | **−0.1800** | **0.000076** |
| Cyanobacteria | −0.00022 | 0.78 | 0.000001 | 0.99 |
| Fusobacteria | −0.00011 | 0.82 | −0.00035 | 0.48 |
| Patescibacteria | 0.00054 | 0.94 | −0.013 | 0.12 |
| Proteobacteria | −0.08200 | 0.53 | −0.2200 | 0.46 |
| **Tenericutes** | 0.04600 | 0.72 | **0.3000** | **0.031** |
| **Verrucomicrobia** | **−0.28000** | **0.023** | 0.00385 | 0.17 |
| Diversity Index | Beta | p-value | Beta | p-value |
| **Shannon** | **0.124** | **0.04** | 0.027 | 0.59 |
| **Chao** | **48.2** | **0.013** | −30.6 | 0.097 |

Relative abundance (%) and diversity indices were compared longitudinally across genotypes using linear mixed effects models and stratified by sex. Betas are the mean difference in genotypes longitudinally across disease course (between 5 and 19 weeks for females and 5 and 16 weeks for males). Phyla and their respected values are bolded where p < 0.05. N = 11–19/genotype/sex.

age, in contrast to the persistence of this phenotype from 6 to 15 weeks in mutant males (Fig. 1b). The differences between overlap distances in *Mecp2* mutant compared to *Mecp2* wild-type littermates were also greater in males, specifically 0.93 ± 0.18 cm at 13 weeks of age compared to 0.54 ± 0.16 cm at 16 and 17 weeks of age in females. In addition, stride length and distance between hind leg placement was also impaired in *Mecp2-e1* mutant mice, demonstrating earlier onset and severity in males compared to females (Supplementary Fig. 1).

While neurological and motor phenotypes presented earlier and with greater severity in *Mecp2-e1*$^{-/y}$ males, *Mecp2-e1*$^{-/+}$ females exhibited earlier onset of progressive body weight gain throughout disease course than males (Fig. 1c). Longitudinally, female *Mecp2-e1*$^{-/+}$ mice weighed more than *Mecp2-e1*$^{+/+}$ littermates, and a non-significant trend was observed in the same increased direction for *Mecp2-e1*$^{-/y}$ males ($p = 0.058$). *Mecp2-e1*$^{-/+}$ females began to significantly increase in body weight at 7 weeks of age and continued to gain weight through 19 weeks of age, with female *Mecp2-e1*$^{-/+}$ mice weighing an average of 13.5 ± 2.8 grams more than *Mecp2-e1*$^{+/+}$ littermates. There were not any statistically significant differences in body weight between *Mecp2-e1*$^{-/y}$ and *Mecp2-e1*$^{+/y}$ males at any age. Together, these longitudinal phenotyping analyses complement previous studies in the disease relevant *Mecp2-e1* RTT mouse model by demonstrating progressive yet distinct onset differences in neurological, motor, and metabolic phenotypes between sexes[38,40].

**Mice deficient in MeCP2-e1 display sex-specific alterations in the fecal microbiome beginning early in postnatal life.** To identify microbial signatures of RTT disease progression in the gastrointestinal tract, we carried out and analyzed 16 S sequencing on fecal samples collected longitudinally on a weekly basis between 5 and 19 weeks of age for female *Mecp2-e1*$^{-/+}$ and *Mecp2-e1*$^{+/+}$ littermates and between 5 and 16 weeks of age for male *Mecp2-e1*$^{-/y}$ and *Mecp2-e1*$^{+/y}$ mice. Illumina MiSeq sequencing reads were resolved to amplicon sequence variants (ASVs) and classified by Phylum, Class, Order, Family, Genus, and Species. Rarefaction curves indicated that at a 250 ASVs detected cut-off, the majority of diversity in the population had been sampled (Supplementary Fig. 2); only 6 of 787 total samples did not have at least 250 ASVs and were removed from downstream analysis.

In general, ASVs identified from all fecal samples were primarily from the *Firmicutes* and *Bacteroidetes* phyla, which together made up over 95% of ASVs, with the remainder lower relative abundance phyla made up of *Actinobacteria*, *Cyanobacteria*, *Fusobacteria*, *Patescibacteria*, *Proteobacteria*, *Tenericutes*, and *Verrucomicrobia* (Supplementary Fig. 3), consistent with the typical adult mouse gut microbiome[45]. However, mutant *Mecp2-e1* mice had sex-specific, differential relative abundances of *Firmicutes*, *Bacteroidetes*, *Actinobacteria*, *Tenericutes*, and *Verrucomicrobia* longitudinally across disease course (Table 1). Longitudinally, female *Mecp2-e1*$^{-/+}$ mutants had increased relative abundance of Firmicutes by 3.7% and decreased relative abundance of *Bacteroidetes* by 3.4% compared to *Mecp2-e1*$^{+/+}$ littermates. In contrast, males had the opposite genotype-related shifts in relative *Firmicutes* and *Bacteroidetes* abundance, specifically a decreased relative abundance of Firmicutes by 5.1% and an increased relative abundance of Bacteroidetes by 5.2% in *Mecp2-e1*$^{-/y}$ versus *Mecp2-e1*$^{+/y}$ littermates. Notably, the expansion of Firmicutes observed in female *Mecp2-e1*$^{-/+}$ mutants was due to increased relative abundance of *Clostridia* (Supplementary Data 1), a class of microbes that regulate the gut-brain axis[46]. In addition, female *Mecp2-e1*$^{-/+}$ mutants had 0.28% lower relative abundance of *Verrucomicrobia* than *Mecp2-e1*$^{+/+}$ littermates, while male *Mecp2-e1*$^{-/y}$ mutants had 0.18% lower relative abundance of *Actinobacteria* and 0.30% higher relative abundance of *Tenericutes* compared to *Mecp2-e1*$^{+/y}$ littermates. In addition to sex-specific relationships between genotype and relative abundance of various microbial Phyla longitudinally, there were also several significant relationships cross-sectionally (Supplementary Fig. 4).

Microbiome diversity is often considered an indicator of gut health, and studies have suggested that the fecal microbiome is less diverse in RTT patients[34,35]. Surprisingly, we found that in females, two measures of microbiota diversity, the Shannon and Chao indices, were increased longitudinally across disease progression in mutant *Mecp2-e1* mice relative to wild-type littermates (Table 1). However, there were no significant differences in diversity scores between mutant and wild-type males. Cross-sectionally, the most pronounced differences in microbiota diversity scores were at 12 and 16 weeks in *Mecp2-e1*$^{-/+}$ females (Supplementary Fig. 5). Despite the lack of statistically significant differences in longitudinal diversity scores in *Mecp2-e1*$^{-/y}$ males, cross-sectional analyses identified significantly different diversity at 5, 6, and 8 weeks of age between *Mecp2-e1*$^{-/y}$ and *Mecp2-e1*$^{+/y}$ littermates (Supplementary Fig. 5).

To understand individual microbial perturbations across RTT disease progression, we investigated the relationships between each ASV and *Mecp2-e1* genotype in both male and female mice using limma voom[47]. Longitudinally, across disease progression, there was a similar number of ASVs that were associated with the main effect of genotype in both females (406) and males (433) (Fig. 2a). Only ~25% (114) of genotype-associated ASVs in *Mecp2-e1*$^{-/+}$ females were also associated with *Mecp2-e1* genotype in males, while 292 genotype-associated ASVs were identified in females versus 319 genotype-associated ASVs detected in *Mecp2-e1*$^{-/y}$ males only (Supplementary Data 2 & 3). Individual ASV and *Mecp2-e1* genotype associations were significant as early as 5 weeks of age for both females and males, with the largest number of associations at 12 and 16 weeks in females and at 14 and 15 weeks in males (FDR < 0.05; Fig. 2b). To identify dynamic changes in gut microbiota phyla across RTT disease progression, we identified ASVs with a significant genotype by age interaction, and found 591 in females compared with 585 in males (Supplementary Data 4 & 5). Visualization of the top 25 ASVs with genotype by age interactions within the fecal microbiota of *Mecp2-e1* mutant mice compared to wild-type

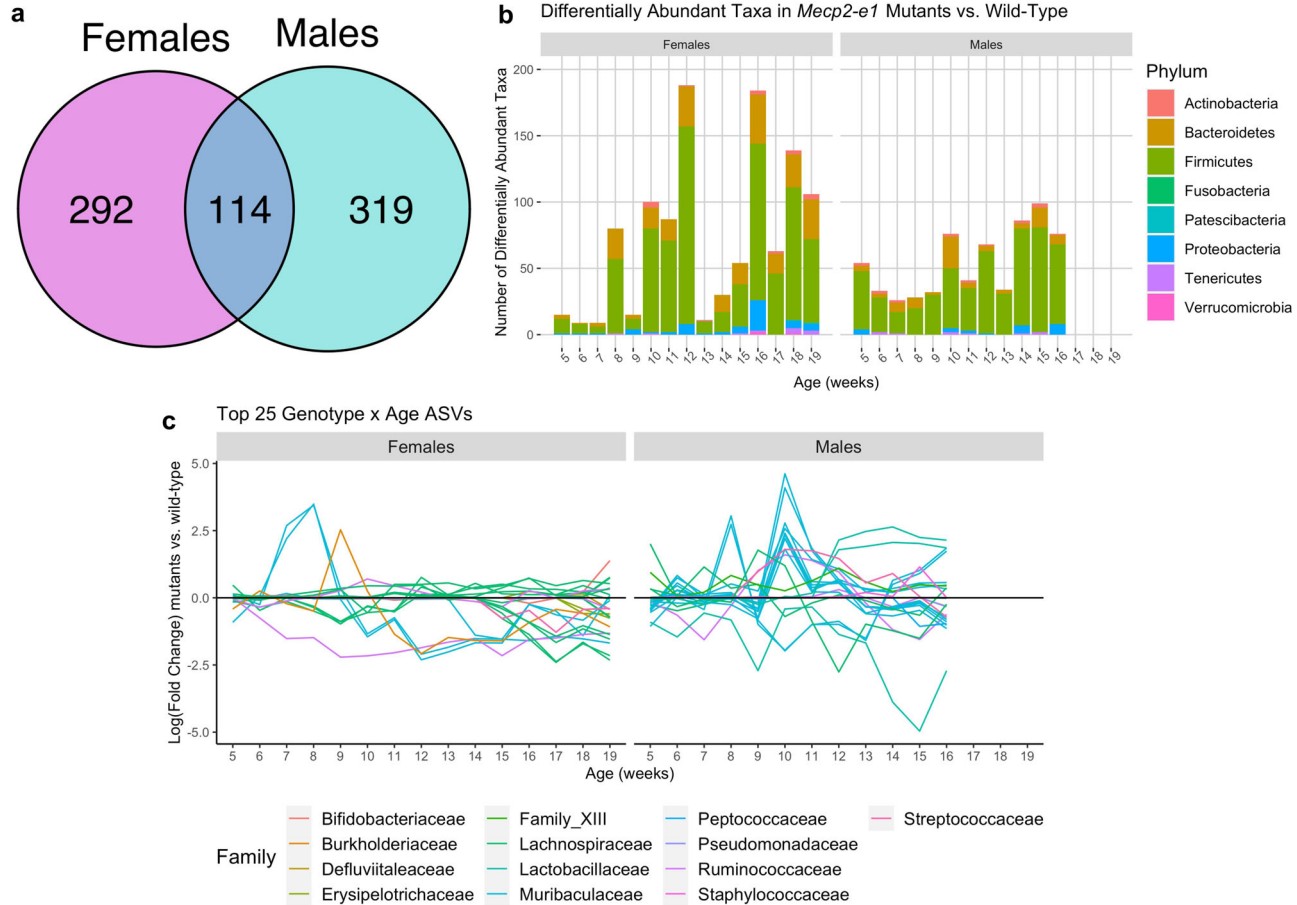

**Fig. 2 Longitudinal differences in the fecal microbiome in mutant vs. wild-type *Mecp2-e1* mice. a** Venn diagram comparing ASVs that were significantly associated (FDR < 0.05) with genotype longitudinally across disease course in females vs. males, while controlling for within-litter effects. **b** Number of differentially abundant taxa (ASVs) by genotype in females and males cross-sectionally at each time point between 5 and 19 weeks of age in females and 5 and 16 weeks of age in males, colored by phylum (FDR < 0.05). **c** the top 25 ASVs with the most statistically significant genotype by age interaction plotted by log(fold change) in mutants vs. wild-type (wt) mice at each time point between 5 and 19 weeks of age in females and 5 and 16 weeks of age in males. Each line represents a single ASV and lines are colored by Family. N = 11–19/genotype/sex.

littermates across time, demonstrate the dynamic nature of the gut microbiome composition throughout disease progression (Fig. 2c). Families representing the top 25 genotype by age ASVs in both sexes were *Lachnospiraceae*, *Muribaculaceae*, *Ruminococcaceae*, *Streptococcaceae*, and *Burkholderiaceae*, while those exclusive to females were *Erysipelotrichaceae*, *Defluviitaleaceae*, and *Bifidobacteriaceae*, and those exclusive to males were *Lactobacillaceae*, *Family_XIII*, *Pseudomonadaceae*, *Staphylococcaceae*, and *Peptococcaceae*. Together, this data demonstrate that the gut microbiome is dynamically altered during disease progression in both female and male *Mecp2-e1* mutants, however there are distinct sex-specific signatures of specific families and ASVs.

**Fecal microbiota alterations in Mecp2-e1 mutant mice precede and reflect longitudinal neurological, motor, and metabolic phenotypes.** To test the hypothesis that specific microbiota may impact RTT disease progression, each neurological, motor, and metabolic phenotype score was tested for association with each genotype-associated ASV. In females and males, 110 and 90 ASVs were associated with neurophenotyping score, 33 and 47 ASVs with gait, and 81 and 103 ASVs with body weight, respectively (Supplementary Data 6-11). Figure 3 shows the RTT phenotype-associated ASVs that were persistently altered by genotype at multiple different time points throughout disease course,

specifically those significantly associated with at least one phenotype, as well as with *Mecp2-e1* genotype in at least 25% of the time points measured (four time points in females, three in males). Many ASVs associated with both body weight and genotype in females, including *Rikenellaceae Alistipes*, *Lachnospiraceae NK4A136 group*, *Lachnospiraceae UCG-006*, *Ruminococcaceae Oscillibacter*, *Peptostreptococcaceae Romboutsia*, and *Ruminococcaceae Ruminococcus 1* genera (Fig. 3a). Persistent genotype-related changes in these ASVs began concurrently with significant genotype-related body weight changes at 7 weeks of age. In contrast to *Mecp2-e1* mutant females, males had far fewer ASVs that were associated with both body weight and genotype persistently (three compared to 10; Fig. 3b). ASVs significantly associated with neurophenotyping score and genotype in females included *Ruminococcaceae Candidatus Soleaferrea*, *Ruminococcaceae Oscillibacter*, *Peptostreptococcacea Romboutsia*, and *Lachnospiraceae Roseburia* genera (Fig. 3c). Remarkably, significant genotype-related changes in *Romboutsia* and *Candidatus Soleaferrea* were observed as early as 8 weeks, which was two weeks prior to significant neurophenotyping scores in *Mecp2-e1* mutant females. In males, neurophenotype associated ASVs distinctly belonged to the genera *Bifidobacteriaceae Bifidobacterium*, *Erysipelotrichaceae Faecalibaculum*, and *Ruminococcaceae Ruminiclostridium 9* (Fig. 3d). Similar to females, *Mecp2-e1* mutant males also demonstrated significant genotype-related differences in these ASVs prior to onset of neurophenotyping score deficits in mutant

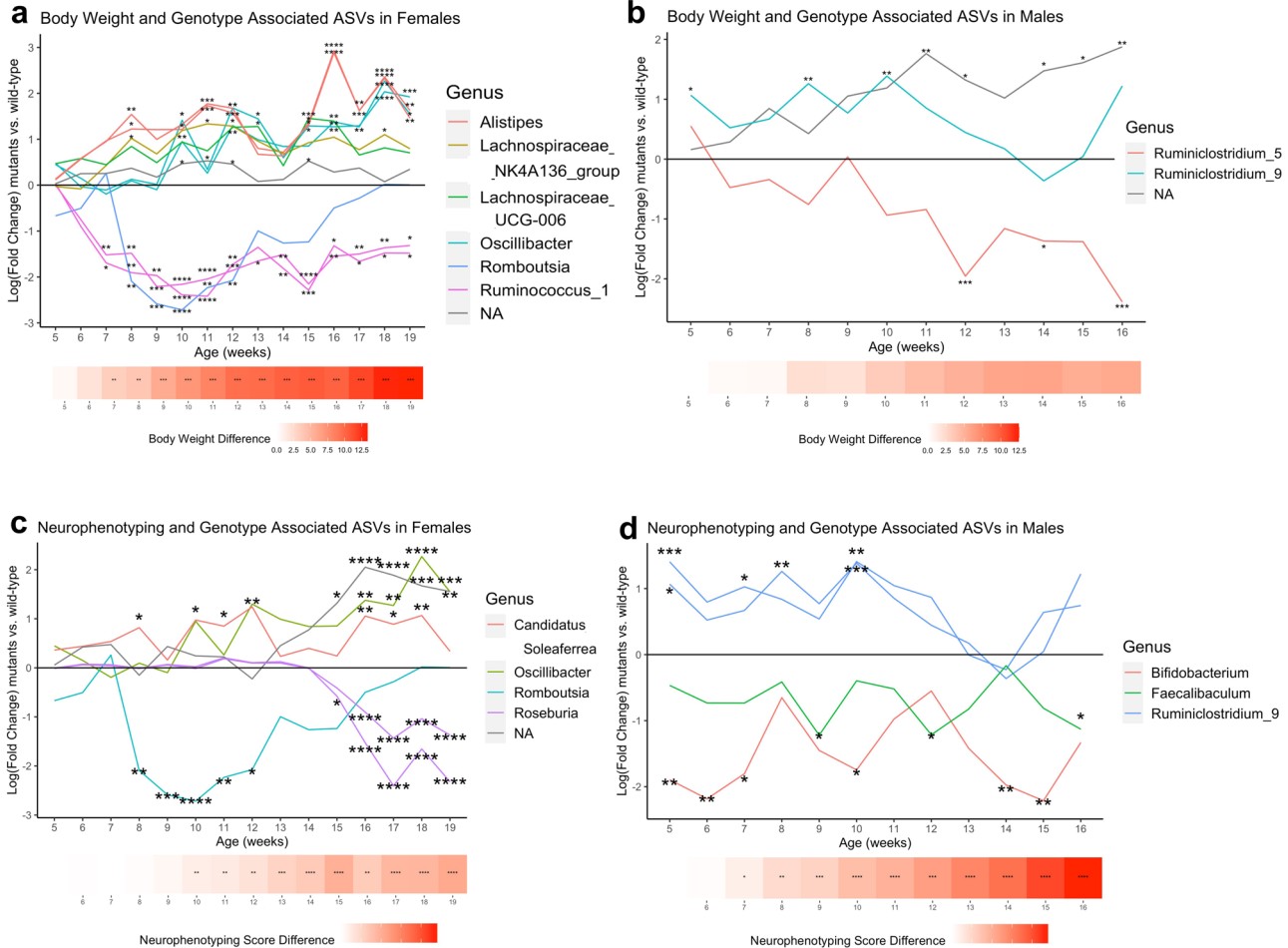

**Fig. 3 Differences in genotype- and phenotype-associated ASVs in mutant vs. wild-type *Mecp2-e1* mice across disease course.** Log(fold change) in normalized read counts of mutant vs. wild-type (wt) mice at each time point in ASVs that were significantly associated with phenotype and genotype at a minimum of one-quarter of the time points in both females and males. Each line represents an individual ASV and lines are colored by Genus. Heatmaps at the bottom of each graph are colored red with increasing intensity based on the difference in phenotype measure by genotype at each time point as reported in Fig. 1. **a** ASVs associated with both body weight and genotype in females, **b** ASVs associated with both body weight and genotype in males, **c** ASVs associated with both neurophenotyping score and genotype in females, **d** ASVs associated with both neurophenotyping score and genotype in males. $N = 11–19$/genotype/sex. *FDR < 0.05, **FDR < 0.01, ***FDR < 0.001 in *Mecp2-e1* mutant vs. wild-type mice cross-sectionally at each time point.

mice. We also observed a small number of ASVs associated with both gait and genotype persistently in females and males (two and one, respectively), and genotype-related changes in these ASVs were observed concurrently with onset of gait abnormalities in females but after onset of gait abnormalities in males (Supplementary Fig. 6).

**Gut inflammation and motility are impacted in Mecp2-e1 mutants in a sex-specific manner.** An altered gut microbiome may be associated with gut inflammation and impaired motility in *Mecp2-e1* mutant mice. Thus, we assessed longitudinal gut inflammation by measuring cytokine levels in fecal samples from wild-type and mutant *Mecp2-e1* mice at 6, 10, and either 14 (males) or 18 weeks of age (females). We found that female and male mutant mice had distinctly altered fecal cytokine profile across disease progression. While decreased fecal interferon-gamma (IFNγ) levels were observed in both male and female mutant mice, interleukin-4 (IL-4) was higher in female mutants but lower in male mutants relative to wild-type (Fig. 4a, b; Supplementary Data 12). Furthermore, genotype-related changes in IL-4 and IFNγ were present throughout disease progression beginning as early as 6 weeks in males. In mutant females, lower

IFNγ levels were not observed until 10 weeks and higher IL-4 levels were only observed in later stages of disease progression at 18 weeks. Since IFNγ is mainly associated with T helper 1 (Th1) cell and IL-4 associated with T helper 2 (Th2) cell immune responses, these results indicate that female *Mecp2-e1* mutants exhibited a shift towards a Th2 response at 18 weeks of age. In addition, longitudinally, female mutants exhibited increased fecal IL-1α, a pro-inflammatory cytokine, compared to wild-type controls, however there were no differences between mutant and wild-type IL-1α levels in males (Fig. 4c). *Mecp2-e1* mutant males also had decreased fecal IL-17 levels across disease course compared to wild-type littermates (Supplementary Fig. 7, Supplementary Data 12). In females, IL-17 was mostly absent with few samples with detectable IL-17 in females. *Mecp2-e1* mutant females also had decreased keratinocytes-derived chemokine (KC) levels compared to wild-type littermates at 10 weeks of age and increased relative KC levels at 18 weeks of age. In males there were few samples with detectable levels of KC (Supplementary Fig. 7, Supplementary Data 12).

In addition to gut inflammatory cytokines, we also assessed gut motility by measuring the number of fecal pellets passed by each mouse within a five-minute period. Both female and male mutants passed more fecal pellets than wild-type controls

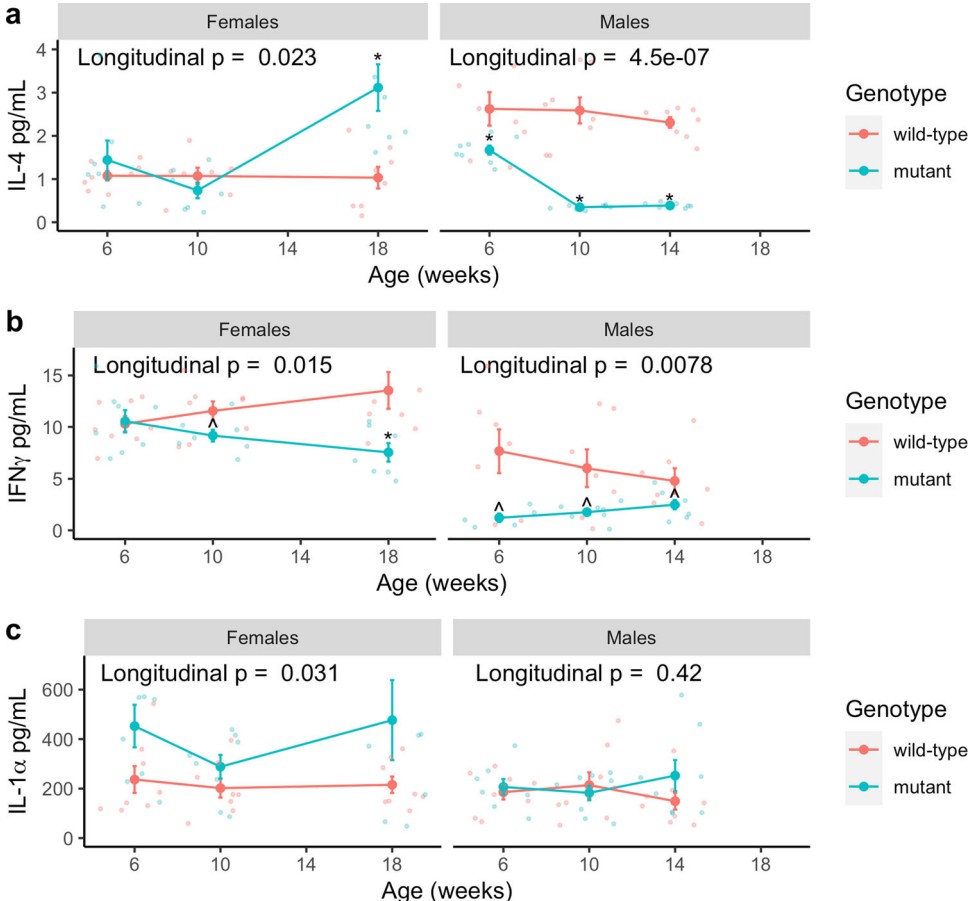

**Fig. 4 Fecal levels of cytokines in *Mecp2-e1* mutant vs. wild-type mice across disease course.** Comparison of **a** fecal interferon-gamma (IFNγ), **b** interleukin-4 (IL-4), and **c** interleukin-1alpha (IL-1α) levels between *Mecp2-e1* mutant and wild-type (wt) mice longitudinally and cross-sectionally across disease course (at 6, 10, and 18 weeks in females and 6, 10, and 14 weeks in males). Analyses were stratified by sex and controlled for within-litter effects. N = 6-8/genotype/sex. Detailed Ns for each time point are included in Supplementary Data 12. Each solid dot represents the mean of each group at each time point and error bars are standard error of the mean (SEM). ^FDR < 0.10, *FDR < 0.05 in mutant vs. wild-type controls cross-sectionally.

longitudinally, though males did not have any significant differences between mutant and wild-type cross-sectionally (Supplementary Fig. 8). Female *Mecp2-e1* mutants displayed increased fecal pellet passage as early as 6 weeks of age, but there was not a consistent difference between mutant and wild-type littermates at all time points (Supplementary Fig. 8).

**Females exhibit stronger associations between fecal metabolites and Mecp2-e1 genotype than males despite heterozygosity.** Since microbial communities in the gut produce a variety of key metabolites, such as short-chain fatty acids (SCFA) relevant to nervous system metabolism, we tested the hypothesis that fecal metabolites may also precede disease phenotypes through longitudinal comparisons of *Mecp2-e1* mutant and wild-type control littermates by measuring fecal metabolites at 5, 9, 16 (males), and 19 (females) weeks of age. We found that *Mecp2-e1*$^{-/+}$ female mutants, but not male mutants, had significantly different levels of fecal SCFAs across disease course compared to wild-type littermates (Fig. 5a; Supplementary Fig. 9). Specifically, female *Mecp2-e1*$^{-/+}$ mutants had higher fecal levels of butyrate, isovalerate, and propionate than wild-type females (FDR < 0.05), with genotype-related differences in butyrate levels that became evident as early as 5 weeks of age, and the most pronounced differences in SCFAs occurring at 9 weeks of age (Fig. 5a; Supplementary Fig. 9).

Striking, sex-specific alterations were also identified through untargeted fecal metabolomics analysis of biogenic amines and lipids in mutant *Mecp2-e1* mutant mice longitudinally. *Mecp2-e1* genotype was associated with a larger number of both fecal biogenic amines and lipids longitudinally in *Mecp2-e1*$^{-/+}$ females compared to *Mecp2-e1*$^{-/y}$ males (Fig. 5b, c). In females, 900 biogenic amines showed a main effect of *Mecp2-e1* genotype, with 792 of those also showing an association with time and/or an interaction between genotype and time (FDR < 0.05) (Fig. 5b). Similar to biogenic amines, females also displayed stronger associations between *Mecp2-e1* genotype and fecal lipids than males. In *Mecp2-e1*$^{-/+}$ females, 93 lipids were associated with genotype longitudinally, and of those, 66 were also associated with time and/or an interaction between genotype and time, further illustrating the dynamic relationship between *Mecp2-e1* and the fecal metabolome across disease progression (Fig. 5c).

Biogenic amines and lipids that could be accurately identified as known compounds and annotated to metabolite databases (89 and 169, respectively) were used in enrichment testing using ChemRICH[48] to identify clusters of related metabolites that were longitudinally associated with *Mecp2-e1* genotype. There were 15 clusters of metabolites associated with genotype in *Mecp2-e1*$^{-/+}$ females (FDR < 0.05; Table 2, Supplementary Fig. 10), but none in *Mecp2-e1*$^{-/y}$ males. The top five clusters of fecal metabolites whose levels were altered in *Mecp2-e1*$^{-/+}$ vs. *Mecp2-e1*$^{+/+}$ littermates were unsaturated triglycerides, dipeptides, phosphatidylcholines,

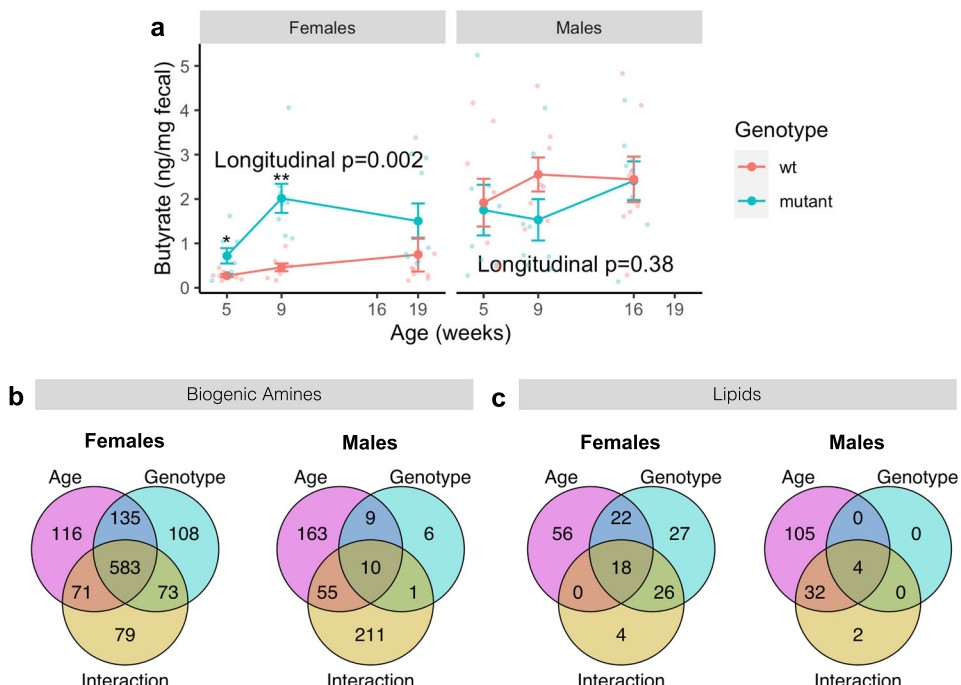

**Fig. 5 Differentially abundant fecal metabolites by *Mecp2-e1* genotype and age in female and male mice. a** Difference in butyrate concentrations in fecal samples in *Mecp2-e1* mutant vs. wild-type (wt) mice longitudinally and cross-sectionally across disease course (at 5, 9, and 19 weeks of age in females and 5, 9, and 16 weeks of age in males), stratified by sex and controlling for within-litter effects. Each solid dot represents the mean of each group at each time point and error bars are standard error of the mean (SEM). $N = 8$/genotype/sex. *FDR < 0.05, **FDR < 0.01 in *Mecp2-e1* mutant vs. wild-type cross-sectionally. Longitudinal *p*-values represent the overall association between mutant and wild-type mice across disease course using linear mixed effects models. **b** Venn diagrams comparing fecal biogenic amines and **c** comparing fecal lipids that were longitudinally (at 5, 9, and 19 weeks for females and 5, 9, and 16 weeks for males) associated with age, genotype, and a genotype by age interaction (FDR < 0.05) in both females and males. $N = 8$/genotype/sex.

**Table 2 Fecal metabolite clusters significantly altered (FDR < 0.05) in *Mecp2-e1*$^{-/+}$ vs. *Mecp2-e1*$+/+$ females.**

| Cluster name | Num. analytes/ cluster | p-value | FDR | Key compound | Num. altered metabolites | Num. increased | Num. decreased |
|---|---|---|---|---|---|---|---|
| Unsaturated triglycerides | 39 | 1.5E-16 | 3.7E-15 | Triglyceride 52:1 | 26 | 8 | 18 |
| Dipeptides | 4 | 2.9E-08 | 2.3E-07 | Ala-Ala | 4 | 3 | 1 |
| Phosphatidylcholines | 26 | 3.4E-08 | 2.3E-07 | Phosphatidylcholine 32:0 | 10 | 5 | 5 |
| Amino acids, basic | 5 | 3.8E-08 | 2.3E-07 | N-Methyltyrosine | 5 | 4 | 1 |
| Unsaturated fatty acids | 24 | 0.000036 | 0.00017 | Fatty Acid 18:2 | 10 | 5 | 5 |
| Indoles | 3 | 0.0001 | 0.0004 | N-.alpha.-Acetyl-L-arginine | 3 | 1 | 2 |
| Azoles | 3 | 0.00015 | 0.0005 | Creatinine | 2 | 2 | 0 |
| Phosphatidylethanolamines | 8 | 0.00034 | 0.001 | Phosphatidylethanolamine 34:1 | 3 | 2 | 1 |
| Pyridines | 8 | 0.00084 | 0.0022 | 3-Aminopyridine | 4 | 4 | 0 |
| Carnitines | 5 | 0.0013 | 0.0031 | 3-Dehydrocarnitine | 3 | 3 | 0 |
| Unsaturated ceramides | 8 | 0.0023 | 0.005 | Ceramide d39:1 | 5 | 1 | 4 |
| Saturated triglycerides | 3 | 0.0025 | 0.005 | Triglyceride 46:0 | 3 | 0 | 3 |
| Amino acids | 5 | 0.0047 | 0.0087 | Isoleucine | 3 | 3 | 0 |
| Diglycerides | 8 | 0.021 | 0.037 | Diglyceride 38:5 | 4 | 3 | 1 |
| Saturated FA | 18 | 0.031 | 0.05 | Fatty Acid 20:0 | 3 | 3 | 0 |

Chemical enrichment clustering analysis was carried out on known metabolites using CHEMRich to identify clusters of chemically similar fecal metabolites that were significantly altered in mutant (−/+) vs. wild-type (+/+) females longitudinally across disease course (measurements taken at 5, 9, and 19 weeks of age). Cluster size is the number of metabolites in each cluster and altered metabolites are the number of metabolites within that cluster that were altered in mutant vs. wild-type females. The key compound is the metabolite with the lowest *p*-value in each cluster. The number of altered metabolites that were increased and decreased in mutant vs. wild-type females are represented in the adjacent columns. $N = 8$/genotype.

basic amino acids, and unsaturated fatty acids. Unsaturated triglycerides, indoles, unsaturated ceramides, and saturated triglycerides were primarily decreased in fecal matter collected longitudinally from *Mecp2-e1*$^{-/+}$ compared to *Mecp2-e1*$^{+/+}$ female littermates, whereas dipeptides, basic amino acids, azoles, phosphatidylethanolamines, pyridines, carnitines, amino acids,

diglycerides, and saturated fatty acids were primarily increased. There was an equal split between the number of increased and decreased species of phosphatidylcholines and unsaturated fatty acids in female mutants compared to wild-type. The full list of metabolites annotated to each cluster and their respective *p*-values and foldchange values can be found in Supplementary Data 13.

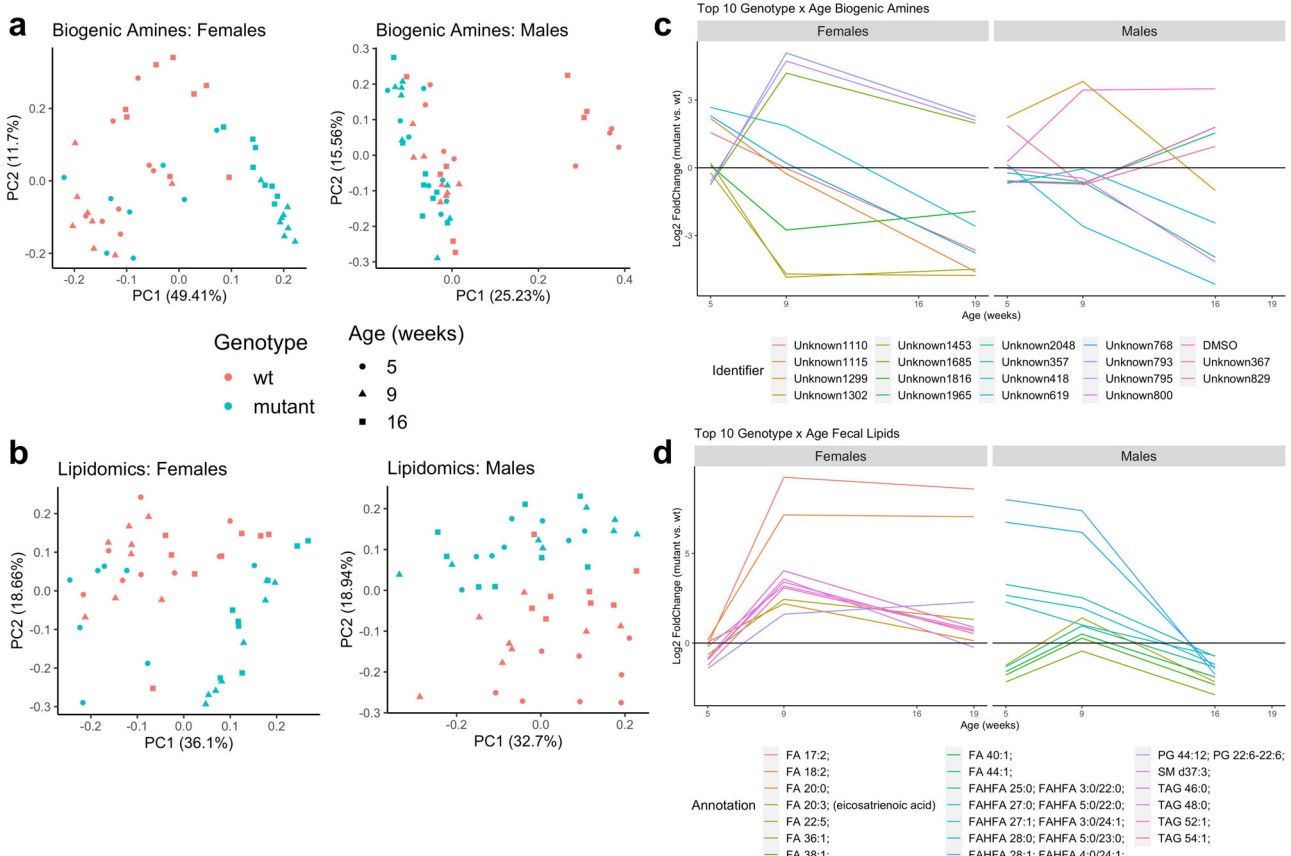

**Fig. 6 Longitudinal fecal metabolome profiles in mutant vs. wild-type *Mecp2-e1* mice.** Principal component analysis (PCA) plots of the top two principal components (PC1 and PC2) for fecal biogenic amines (**a**) and fecal lipids (**b**) with each point representing a single fecal sample and genotype represented by color and time point represented by shape. The log2(fold change) between mutant and wild-type (wt) for the top 10 biogenic amines (**c**) and lipids (**d**) with the most significant genotype by age interactions in females and males, plotted longitudinally across disease course (at 5, 9, and 19 weeks for females and 5, 9, and 16 weeks for males). Each line represents a single biogenic amine or lipid and is colored as such. FA = fatty acid, FAHFA = fatty acid esters of hydroxy fatty acid, TAG = triglyceride, PG = phosphatidylglycerol, SM = sphingomyelin. N = 8/genotype/sex.

**Multi-variable associations of fecal metabolites with Mecp2-e1 genotype, sex, age, phenotype, and microbiome**. To identify shifts in the fecal metabolome over disease course, we carried out a principal component analysis (PCA) and examined longitudinal patterns of metabolites that displayed a strong genotype by age interaction. In *Mecp2-e1*$^{-/+}$ females, PCA plots for both biogenic amines and lipids showed that mutants clustered together with wild-type mice at 5 weeks of age but separated from *Mecp2-e1*$^{+/+}$ controls at 9 weeks and 19 weeks of age (Fig. 6a, b). This indicates that the fecal metabolome underwent a drastic change in female *Mecp2-e1*$^{-/+}$ mutants at approximately 9 weeks of age. *Mecp2-e1*$^{-/y}$ males, on the other hand, did not show distinctive clustering by genotype or age in fecal biogenic amines, and showed clustering by genotype, but not age, in fecal lipidomic measures (Fig. 6a, b). Examination of the top ten fecal biogenic amines and lipids with a significant genotype by age interaction further demonstrates that 9 weeks is a critical age at which the fecal metabolome in *Mecp2-e1*$^{-/+}$ female mutants diverged from wild-type *Mecp2-e1*$^{+/+}$ littermates (Fig. 6c, d). Furthermore, fecal lipids displayed a distinctive pattern in *Mecp2-e1*$^{-/+}$ females compared to controls, indicating a relative increase of fecal lipids in mutants across disease progression. In contrast, the top ten biogenic amines and lipids with a genotype by age interaction identified in males showed differences in biogenic amines progressively throughout the disease course, whereas lipids displayed the largest differences at 5 weeks (Fig. 6c, d). Notably, the biogenic amines with the most significant genotype by age interaction in both *Mecp2-e1* mutant males and females were metabolites that were

unable to be identified (unknowns). The most dynamic lipids across disease course were primarily long chain saturated and unsaturated fatty acids (LCFAs) and triglycerides in *Mecp2-e1*$^{-/+}$ females, but in *Mecp2-e1*$^{-/y}$ males, they were primarily very long chain fatty acids (VLCFAs) and fatty acid esters of hydroxy fatty acids (FAHFAs). Together, these data demonstrate a female-specific dynamic shift in the fecal metabolome in *Mecp2-e1*$^{-/+}$ mutant mice occurring as early as 5 weeks of age for butyrate and 9 weeks of age for lipids and biogenic amines.

In order to integrate the fecal metabolome with neurometabolic phenotypes and the microbiome, we applied a weighted gene co-expression network analysis (WGCNA) approach to metabolomic data. WGCNA reduced the complexity of individual metabolites into modules of highly correlated metabolites that were associated with genotype, age, fecal microbiota, and phenotypic traits. WGCNA analysis identified 18 fecal metabolite modules in females and 33 fecal metabolite modules in males (Supplementary Data 14 & 15). Modules were designated according to a hub metabolite, and metabolites that did not group into a network were placed into the gray unassigned module. Of the identified modules, five were associated with *Mecp2-e1* genotype in females and six were associated with *Mecp2-e1* genotype in males (Supplementary Fig. 11a, 11b). In females, genotype-associated metabolite modules were also related to age (three modules), fecal butyrate levels (two modules), gait as measured by overlap distance (three modules), body weight (three modules), neuro-phenotyping score (three modules), fecal IL-4 (one module), and

fecal IFNγ (one module). *Mecp2-e1* genotype-associated metabolite modules in males showed a less significant correlation with these measures; only one of six modules was associated with gait, one was associated with neuro-phenotyping score, and three were associated with fecal IL-4, while there were none associated with age, fecal butyrate levels, fecal IFNγ levels, or body weight. These data provide further evidence that fecal metabolites may be sensitive indicators of RTT progression, especially in females.

*Mecp2-e1* genotype-associated fecal metabolite modules were also correlated with numerous fecal microbiota in both females and males. In females, *Mecp2-e1* genotype-associated metabolite modules had a significant relationship with 28 ASVs (Supplementary Fig. 11c), and in males, *Mecp2-e1* genotype-associated metabolite modules were associated with 56 ASVs (Supplementary Fig. 11d). Of these, 10 were also associated with gait, neuro-phenotyping score, and/or body weight in females, and 14 were also associated with at least one of these phenotypes in males, demonstrating a high degree of intersection between *Mecp2-e1* genotype, the fecal microbiome, the fecal metabolome, and disease phenotype. The ASVs that were associated with genotype-related metabolite modules were from similar families across males and females, predominantly from the *Lachnospiraceae* and *Ruminococcaceae* families. Notably, one ASV, *Ruminococcaceae Oscillibacter*, was associated with two metabolite modules and all three disease phenotypes in females, and one ASV belonging to the *Lachnospiraceae* family of unknown genus was associated with two metabolite modules and all three disease phenotypes in males. The high degree of intersectionality between *Mecp2-e1* genotype, the fecal microbiome, the fecal metabolome, fecal cytokines and disease phenotypes suggest a significant role for the GI tract in RTT disease progression.

**Female mice deficient in MeCP2-e1 have a distinctive brain lipid signature that reflects genotype-related alterations in fecal lipids.** The striking patterns of increased fecal lipids, including SCFAs, in *Mecp2-e1^{−/+}* females compared to controls across disease course indicate potential lipid malabsorption in the GI tract, which could impact the lipid composition of the brain. Furthermore, previous studies have found that lipid metabolism is disrupted in the brain of *Mecp2* null mice[26,49,50]. Thus, we tested the hypotheses that the lipidome was altered in the cortex from *Mecp2-e1* mutants and correlated with the fecal metabolic changes. PCA plots of brain lipids supported this hypothesis and demonstrate a clear separation of *Mecp2-e1^{−/+}* females from *Mecp2-e1^{+/+}* littermates (Fig. 7a). In contrast, males did not cluster separately by *Mecp2* genotype for brain lipid composition. There were 35 lipids that differed by genotype in female brain at *p*-value <0.05, although these were no longer significant at FDR 0.05 (Supplementary Data 16). Enrichment analysis of these lipids indicated that there were two clusters of down-regulated lipids in the cortex from mutant females compared to wild-type; one cluster of phosphatidylethanolamines and one of sphingomyelins (Supplementary Data 17). A full list of the lipids in each cluster can be found in Supplementary Data 18. Unsaturated phosphatidylcholines also trended towards being depleted in *Mecp2-e1^{−/+}* females. In contrast, there were only five lipids altered in the cortex of *Mecp2-e1^{−/y}* compared to *Mecp2-e1^{+/y}* male littermates at *p* < 0.05 (Supplementary Data 19), and there were no significantly enriched lipid clusters in males.

If lipid malabsorption in the GI tract influences lipid composition of the brain in RTT females, then fecal lipid levels should be associated with brain cortical lipids. In support of this hypothesis, we found a high degree of correlation between the top 10 genotype-associated fecal lipids at 9 weeks of age and the top 10 genotype-associated cortical lipids at 19 weeks of age in

females (Fig. 7b). These correlations were predominantly in the inverse direction, indicating that lipids that were increased in fecal matter were associated with decreased lipids in the brain cortices. Many of the lipids that were decreased in *Mecp2-e1^{−/+}* cortex relative to *Mecp2-e1^{+/+}* female littermates are critical for neuronal function, such as phosphatidylethanolamines, sphingomyelins, and phosphatidylcholines[51–53]. Relative decreases in the cortical levels of these important lipids showed especially strong correlations with relative increases in fecal levels of fatty acid 18:4 and 18:2 (Fig. 7b). Thus, these data suggest that decreased absorption of lipids in the GI tract may negatively impact brain lipids in females with RTT.

## Discussion

This study, is the first of its kind to longitudinally characterize metabolite, microbial, and neuromotor profiles of RTT disease progression in a patient-relevant *Mecp2-e1* mutant mouse model which provided critical insights into pathophysiology. The innovative nature of this study exploits longitudinal measurements in the same mice over disease course which allowed us to (1) identify molecular pathways that emerge prior to overt neuromotor phenotypes, (2) integrate multiomic data to identify connections between neuromotor and metabolic pathways in RTT, and (3) utilized bioavailable tissues (i.e., fecal samples) that can be obtained in human studies, allowing for direct comparisons between this study and those carried out in RTT patients.

We found a high degree of correlation between the gut microbiome, GI tract metabolism, and neuromotor phenotypes longitudinally across disease course, suggesting a role for the gut microbiome and metabolism in RTT disease progression, and uncovering critical disease-relevant microbiota and metabolites. Our findings revealed novel molecular signatures of gut microbial communities and metabolism that emerged prior to onset of neurological and motor phenotypes. Furthermore, this study uniquely identified the timing of molecular phenotypes in heterozygous *Mecp2-e1^{−/+}* females but not hemizygous *Mecp2-e1^{−/y}* males, demonstrating that the intermediate phenotypes of RTT disease progression are sexually dimorphic (Fig. 8). In females, *Mecp2-e1^{−/+}* mutants displayed alterations in fecal microbiota and metabolites as early as 5 weeks of age, prior to onset of disease phenotypes. The earliest disease phenotype that appeared in *Mecp2-e1^{−/+}* females was increased body weight at 7 weeks of age. At 9 weeks of age, *Mecp2-e1^{−/+}* females had a major shift in their fecal metabolome, just prior to the emergence of neurological and motor phenotypes at 10-11 weeks of age, and peak differences in the fecal microbiome at 12 weeks of age. In congruence with previous studies of RTT *Mecp2* null mouse models, neuromotor phenotypes emerged much earlier in *Mecp2-e1^{−/y}* males than in *Mecp2-e1^{−/+}* females and significant differences were observed as early as 6 weeks of age, and early morbidity and mortality occurred relatively early in life at 16 weeks of age. Differences in fecal microbiota occurred concurrently with onset of gait and motor phenotypes in *Mecp2-e1^{−/y}* males and progressed throughout disease course. Although *Mecp2-e1^{−/y}* males trended towards an increased body weight compared to *Mecp2-e1^{+/y}* littermates, there were no statistically significant differences in body weight in male mice, in contrast to female mice. Furthermore, *Mecp2-e1^{−/y}* males exhibited very few alterations in fecal metabolites and brain lipids compared to the more pronounced differences observed in *Mecp2-e1^{−/+}* females. Taken together, our findings demonstrate a sex-specific RTT disease course and indicate that metabolic abnormalities in RTT precede symptom onset in females, suggesting a significant role of perturbed metabolism in disease pathophysiology and progression.

Two studies have previously demonstrated that female RTT patients have an altered gut microbial community compared to

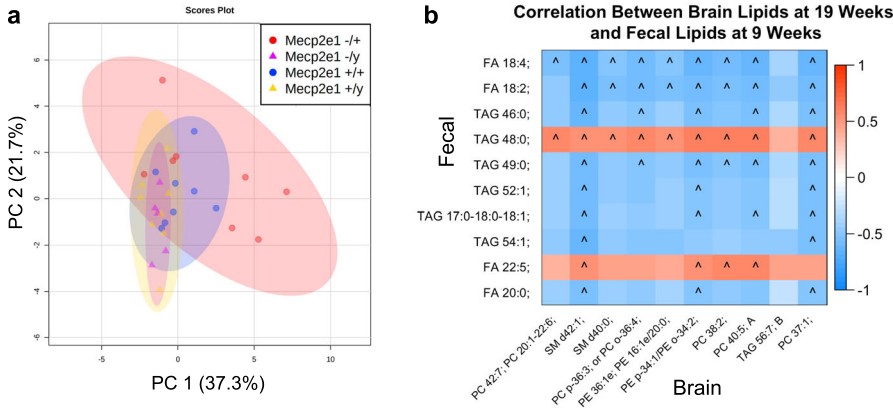

**Fig. 7 Brain lipidome profiles in *Mecp2-e1* mutant and wild-type mice. a** Principal component analysis (PCA) plot of female and male mutant and wild-type cortical lipids measured at the end of disease course (19 weeks for females and 16 weeks for males). The two principal components (PCs) that explain the most variation in samples are plotted with PC 1 on the x-axis and PC 2 on the y-axis. Each dot represents one sample, with colors representing genotype and shapes representing sex. **b** Heatmap depicting the relationship between the 10 fecal lipids at 19 weeks and the 10 brain lipids at 9 weeks with the most significant association with genotype in females. Associations were strongest for fecal lipids at 9 weeks of age, and thus those are depicted here. Blocks are colored based on correlation coefficient, with red representing positive correlations and blue representing negative correlations. FA = fatty acid, FAHFA = fatty acid esters of hydroxy fatty acid, TAG = triglyceride, PG = phosphatidylglycerol, SM = sphingomyelin. $N = 6–8$/genotype/sex. ^FDR < 0.10.

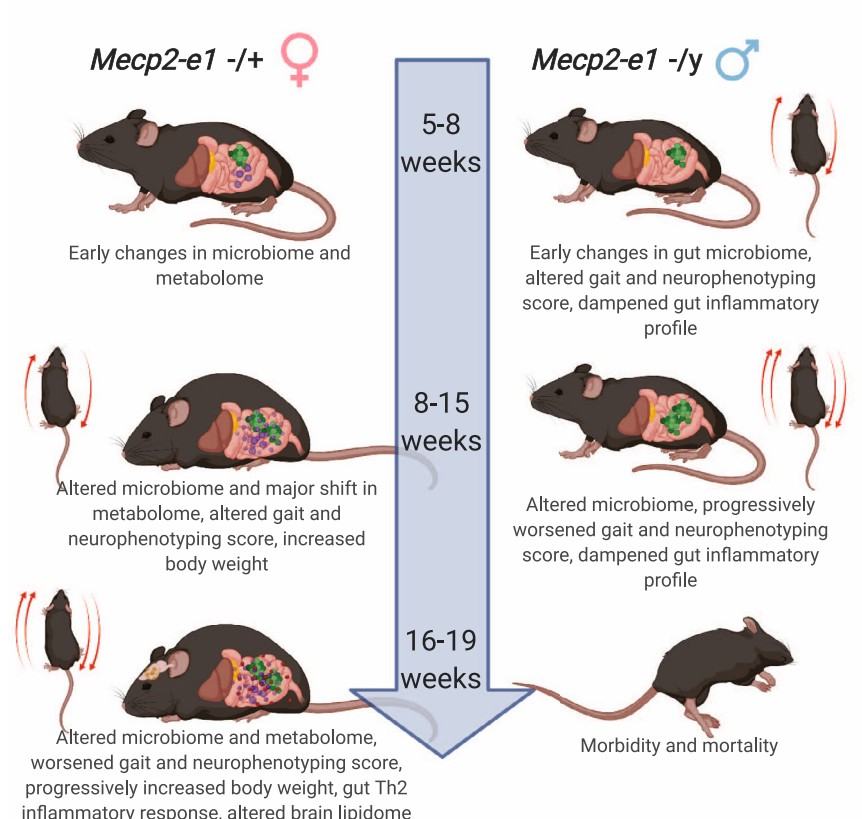

**Fig. 8 Female and male *Mecp2-e1* mutants exhibit divergent phenotypic and molecular RTT disease progression.** In early adulthood at 5–7 weeks of age, *Mecp2-e1* mutant males begin to show declines in motor and neurological function and changes in their gut microbiome. On the other hand, female *Mecp2-e1* mutants at 5–8 weeks of age do not yet display motor and neurological symptoms, but begin to show increased body weight, altered gut microbiota *and* altered gut metabolites, such as short chain fatty acids (SCFAs). Females begin to display progressively increased body weight and drastic shifts in the gut metabolome at 9 weeks of age, prior to persistent declines in neurological and motor function which occurred at 10–11 weeks of age. Peak differences in the gut microbiome occurred at 12 weeks of age in females. Males, on the other hand, display progressive worsening of neuromotor symptoms and progressive changes in the gut microbiome in the absence of drastic shifts in the gut metabolome between 9 and 15 weeks of age. In late-stage RTT disease between 16 and 19 months of age, males exhibit severe morbidity and early mortality, whereas females continue to display progressively increased body weight and neuromotor function, continued evidence of a disrupted gut microbiome and metabolome, a Th2 inflammatory response in the gut, and altered brain lipid profiles. Figure created with BioRender.com.

healthy controls[34,35], but this is the first study to examine gut microbiota in an *Mecp2* deficient mouse model, which has the advantage of controlling diet and isolating the effect of mutations in *Mecp2*. Several of our findings were consistent with those in human RTT patients. For example, our finding in *Mecp2-e1⁻/⁺* females that the reduced fecal abundance of *Bacteroidetes* was concurrent with an increase in *Firmicutes* was consistent with those of Strati et al.[34] in human female patients with RTT, as well as in children with autism spectrum disorder (ASD)[54] and in children with increased body mass index (BMI)[55–57], suggesting a convergent profile reflective of humans with neurodevelopmental and metabolic dysfunction. Notably, *Mecp2-e1⁻/ʸ* mutant males showed the opposite results; *Mecp2-e1⁻/ʸ* mutant males had an increased relative abundance of *Bacteroidetes* and decreased relative abundance of *Firmicutes*, highlighting the sex-specific nature of RTT. Equivalent studies in RTT patients reported increased relative abundance of taxa belonging to the *Clostridia* class[34,35], similar to children with ASD[58], which was consistent with the female-specific increase in relative abundance of *Clostridia* longitudinally in *Mecp2-e1* mutants compared to wild-type control littermates that we observed. Several ASVs that were associated with both genotype, neurological phenotyping score, and body weight were from the *Clostridia* class, including *Oscillibacter* which was significantly and persistently increased in *Mecp2-e1⁻/⁺* females and also associated with neuro-phenotype, gait, and body weight. On the other hand, another member of the *Clostridia* class, *Roseburia*, was decreased both in mutant *Mecp2-e1⁻/⁺* mutant female mice from this study and in female patients with RTT from a human study[35]. We also found that an ASV belonging to the *Roseburia* genus was significantly associated with neurophenotype, though the difference in abundance of *Roseburia* did not emerge until five weeks after genotype-related differences in neurophenotype. Our results were also consistent with the two human RTT studies in the elevated levels of several SCFAs in fecal samples, including propionate, butyrate, and iso-valerate. Of note, propionate and isovalerate are produced by *Clostridia*. In contrast to female mice, male *Mecp2-e1⁻/ʸ* mice did not exhibit consistencies with human RTT microbiome data, and in fact, showed some opposite effects. *Mecp2-e1⁻/ʸ* male mice had decreased *Actinobacteria*, *Erysipelotrichaceae*, and *Bifidobacterium*, whereas they were increased in human RTT patients. There were also some inconsistencies between the fecal microbiome in female *Mecp2-e1⁻/⁺* mutants and human RTT patients. Most notably, *Mecp2-e1⁻/⁺* mutant females had increased richness of the fecal microbiome, but female RTT patients had decreased alpha-diversity scores compared to healthy controls. Cumulatively, our findings experimentally confirmed the relationship between RTT and the gut microbiome from human RTT patients and further demonstrate that *Mecp2* deficient female mice are a better model of human RTT patients than male mice. If this study had only included males, our findings would not have been consistent with microbiome, metabolite, and immune findings from RTT patients. In addition, our study allowed us to identify several taxa that were associated with disease phenotypes and *Mecp2-e1* genotype that warrant further study in their potentially causative role in disease pathogenesis.

Although *Mecp2-e1* mutant females and males exhibited altered gut microbes beginning early in disease progression at 5 weeks of age, the largest numbers of differentially abundant taxa were observed in later stages of disease. In females, the number of differentially abundant taxa fluctuated by over 100 taxa throughout disease course, indicating that the gut microbiome undergoes dynamic changes throughout RTT disease progression in females. The number of differentially abundant taxa peaked at 12 (188 ASVs) and 16 weeks (184 ASVs) in females. The peak differences at 12 weeks may have been a response in the gut

microbiome to onset of neurological and gait abnormalities (at 10 and 11 weeks, respectively), and the peak at 16 weeks coincides with the observed fecal cytokine response, indicative of gut inflammation. In males, on the other hand, there was a gradual, steady increase in the number of differentially abundant taxa with disease progression. The peak occurred at 14 (86 ASVs) and 15 weeks (99 ASVs), but there was a similarly large number of differentially abundant taxa at 16 weeks as well (76 ASVs). Additional studies are needed to elucidate what precisely contributes to the timing of peak differences in gut microbe abundance in RTT.

We found that *Mecp2-e1* mutant females, but not males, exhibited a Th2-type response detectable in the fecal pellet in late stages of disease progression, as demonstrated by increased levels of IL-4 and decreased levels of IFNγ relative to wild-type controls. This is consistent with a clinical study that found increased IL-4 and decreased IFNγ in plasma from RTT patients compared to healthy controls[59]. Interestingly, taken together these data suggest that a Th2 response is a species-independent feature of female MeCP2 deficiency and RTT disease progression. Persistent Th2 responses in the gut have been shown to result in chronic inflammation in other disease processes such as ulcerative colitis[60]. The relative increase in fecal IL-4 levels observed in *Mecp2-e1* mutant females did not emerge until later in RTT disease progression, indicating that gut inflammation may be secondary or in response to alterations in the gut microbiome or metabolism. Further assessment of immune responses in the mucosal layer and at intestinal lymph nodes will be undertaken in follow up studies. In contrast, *Mecp2-e1* mutant males showed relatively decreased levels of fecal IFNγ, IL-4, and IL-17 throughout disease progression. This could indicate a dampened gut immune responsiveness by the complete loss of MeCP2e1 in males that is already present in early stages of disease.

In recent years, metabolic dysfunction has been increasingly recognized as an important component of RTT disease pathology[24]. Our findings that female *Mecp2-e1* mutant mice display progressive obesity and sex-specific alterations in the fecal metabolome corroborate this hypothesis. Importantly, we found that the fecal metabolome showed significant genotype-related differences prior to emergence of neurological and motor phenotypes, and that the fecal metabolome was also highly correlated with these phenotypes. Lipid metabolism, specifically, has been demonstrated to be dysregulated in both RTT patients and mouse models in previously published studies. Blood and plasma metabolites from RTT patients have indicated that cholesterol metabolism, sphingolipid metabolism, and fatty acid metabolism are perturbed in RTT[49,61–65]. Cholesterol and triglyceride metabolism has also been demonstrated to be dysregulated in the liver and brain in mouse models of RTT[26,38,50,66,67]. Our data also support a role for lipid dysregulation in RTT. *Mecp2-e1* mutant females, but not males, had altered fecal levels of many lipid species, including triglycerides, fatty acids, phosphatidylcholines, and phosphatidylethanolamines, beginning at 9 weeks of age, prior to onset of neuromotor phenotypes, suggesting that lipid dysregulation in the GI tract may play a role in disrupting neurological function. Further, the most dynamic genotype-associated fecal lipids were largely increased in female *Mecp2-e1* mutants compared to wild-type littermates, which raises the possibility that deficient lipid absorption in the GI tract contributes to RTT pathology. Phosphatidylethanolamines, phosphatidylcholines, and sphingomyelins were also altered in the cortex of female mosaics, and these brain lipids were inversely associated with fecal lipid levels, providing additional evidence supporting this theory. Notably, the strongest correlations between cortical (19 weeks) and fecal lipid levels were with fecal lipid levels at 9 weeks. Decreased fecal lipid levels in *Mecp2-e1*

mutant females were the most pronounced at 9 weeks but may have resulted in long-term impacts on brain lipid levels. Fecal and cortical lipids were predominantly influenced by *Mecp2-e1* genotype in females, with few effects in males, indicating that the role of lipid dysregulation in the gut is specific to female disease progression in RTT.

Our findings demonstrate that RTT disease progression is sexually dimorphic and suggest that the underlying molecular pathology is inherently different in females than in males. This was surprising given that *Mecp2-e1*$^{-/+}$ mutant females have a wild-type copy of *Mecp2-e1* compared to *Mecp2-e1*$^{-/y}$ males. X-linked dominant human disorders are quite rare, suggesting that the second copy of X-linked genes is usually protective. Most previous studies in animal models of RTT have focused on males hemizygous for mutated or null *Mecp2*, and while critically informative for understanding the function of MeCP2, this resulted in gaps in understanding the complexity of RTT disease progression. Female-specific effects in *Mecp2-e1* mutant mice could be due to sex hormones, cellular mosaicism, or differences in feeding patterns. Sex hormones play a critical role in regulating metabolism[68], and may therefore make females more susceptible to metabolic perturbations by deficits in MeCP2. However, cells that express mutant *Mecp2* have been shown to have non-cell autonomous, or "bad neighborhood" effects, on *MECP2* wild-type expressing cells in mosaic mutant females[15,16,69]. Specifically, these non-cell autonomous effects in astrocytes have been found to dampen dendritic arborization in mouse models[16]. Male, but not female, *Mecp2-e1* mice exhibited significantly reduced food intake and energy expenditure in previous studies[38], which could also contribute to the observed sex-specific effects on the gut microbiome and metabolism. Additional studies on sex hormonal and non-cell autonomous effects in metabolically active tissues such as the liver, adipose, and GI tract are needed to explore these potential mechanisms for sex-specific effects in RTT.

The findings from this study provide insights into the underlying molecular pathways involved in RTT disease progression and demonstrate the sexually dimorphic nature of RTT pathophysiology. We found a convergence of metabolism, gut microbial communities and cytokine profiles, and neuromotor phenotypes in a patient-relevant mouse model of RTT. In females, differences in metabolites and microbes between *Mecp2-e1* mutant and wild-type were observed prior to onset of neuromotor phenotypes, suggesting that these pathways play a role in disease pathology. Critically, our data demonstrated that female *Mecp2-e1* mutant mice recapitulate molecular signatures of human RTT more consistently than male *Mecp2-e1* mutants, including alterations in the gut microbiome, metabolome, and cytokine profiles. We also discovered that lipid malabsorption may contribute to RTT neuromotor phenotypes in females, a potential new pathway that should be studied further for potential therapies. Future animal studies investigating RTT disease pathology and treatments should include females in order to effectively translate findings to human RTT patients.

## Methods

**Mouse breeding and cross fostering.** *Mecp2-e1* mutant and wild-type littermate controls were generated and maintained as described previously[38,40]. Briefly, *Mecp2-e1*$^{-/+}$ mutant heterozygous females (*Mecp2-e1* -/+) were bred with wild-type C57BL/6 J males (Jax 000664) to generate mutant heterozygous female offspring (*Mecp2-e1*$^{-/+}$), mutant hemizygous male offspring (*Mecp2-e1*$^{-/y}$), and wild-type female (*Mecp2-e1*$^{+/+}$) and male (*Mecp2-e1*$^{+/y}$) offspring. Pups were cross-fostered to CD1 foster dams within the first 48 h of birth to prevent the impact of poor maternal care from *Mecp2-e1*$^{-/+}$ mutant dams. After weaning, mice were housed according to sex and genotype. Mice were maintained in a conventional temperature-controlled vivarium on a 12-h light cycle with *ad libitum* access to food and water. All animal experiments were conducted in compliance with the National Institutes of Health Guidelines for the Care and use of Laboratory Animals and were carried out under approval and monitoring by the Institutional Animal Care and Use Committee (IACUC) of the University of California, Davis under IACUC protocol #20621.

**Mouse phenotyping.** Neuro-phenotyping and body weighing were performed weekly at three hours after the beginning of the light cycle (ZT3), as described previously[40]. Briefly, mice were evaluated weekly by the same researcher for ruffled coat, hypoactivity, open skin ulcers, abdomen size and specific responses to tail suspension including hind limb clasping, forelimb 'washing' and side to side flailing. Gait assessment was performed weekly between ZT3 and ZT5 as described previously[38]. Mice were habituated to paint application to feet on the day prior to gait analysis. The fore paws of mice were painted blue and the rear paws were painted red upon removal from home cage. Then mice were placed on blank paper strips in a chamber with a darkened distal end. The mouse was then allowed to walk down a straight alleyway lined with drawing paper. The resulting footprints were analyzed for stride length (distance between successive forelimb and successive hind limb prints), hind-base (distance between the right and left hind prints), front-base (distance between right and left front prints) and paw separation (distance between the forepaw and hind paw placement). For gastrointestinal transit measures, the number of fecal pellets passed within 5 min was counted for each mouse on a weekly basis three hours after the beginning of the light cycle (ZT3).

**Sample collection.** Fecal pellets were collected weekly at 3 h after the beginning of the light cycle (ZT3). Individual mice were placed in a clean cage for 5 min and accumulated fecal pellets were then collected, placed in tubes and frozen on dry ice prior to storage at −80 °C. Cortex samples were collected at 3 h after light cycle start after euthanasia via $CO_2$ at 19 weeks of age for females and 16 weeks of age for males. Cortical samples were flash frozen in liquid nitrogen and stored at −80 °C prior to use.

**Characterization of the fecal microbiome.** Total microbial DNA was isolated from fecal pellets using Qiagen QIAmp PowerFecal DNA kit (Qiagen, Germantown, MD) and measured by spectrophotometry (Nanodrop, Thermo-Fisher, Waltham, MA). Primers 319 F (**TCGTCGGCAGCGTCAGATGTGTATAAGAGACAG**(spacer)*GT*ACTCCTACGGGAGGCAGCAGT) and 806 R (**GTCTCGTGGGCTCGGAGATGTGTATAAGAGACAG**(spacer)CCGGACTACNVGGGTWTCTAAT were used to amplify the V3-V4 domain of the 16 S rRNA using a two-step PCR procedure. In step one of the amplification procedure, both forward and reverse primers contained an Illumina tag sequence (bold), a variable length spacer (no spacer, C, TC, or ATC for 319 F; no spacer, G, TG, ATG for 806 R) to increase diversity and improve the quality of the sequencing run, a linker sequence (italicized), and the 16 S target sequence (underlined). Each 25 µl PCR reaction contained 1 Unit Kapa2G Robust Hot Start Polymerase (Kapa Biosystems), 1.5 mM $MgCl_2$, 0.2 mM final concentration dNTP mix, 0.2 µM final concentration of each primer and 1 µl of DNA for each sample. PCR conditions were: an initial incubation at 95 °C for 3 min, followed by 25 cycles of 95 °C for 45 s, 50 °C for 30 s, 72 °C for 30 s and a final extension of 72 °C for 3 min. In step two, each sample was barcoded with a unique forward and reverse barcode combination using forward primers (**AATGATACGGCGACCACCGAGATCTACAC**NNNNNNNNTCGTCGGCAGCGTC) with an Illumina P5 adapter sequence (bold), a unique 8 nt barcode (N), a partial matching sequence of the forward adapter used in step one (underlined), and reverse primers (**CAAGCAGAAGACGGCATACGAGAT**NNNNNNNNGTCTCGTGGGCTCGG)) with an Illumina P7 adapter sequence (bold), unique 8 nt barcode (N), and a partial matching sequence of the reverse adapter used in step one (underlined). The PCR reaction in step two contained 1 Unit Kapa2G Robust Hot Start Polymerase (Kapa Biosystems), 1.5 mM $MgCl_2$, 0.2 mM final concentration dNTP mix, 0.2 µM final concentration of each uniquely barcoded primer and 1 µl of the product from the PCR reaction in step one diluted at a 10:1 ratio in water. PCR conditions were: an initial incubation at 95 °C for 3 min, followed by 9 cycles of 95 °C for 30 s, 58 °C for 30 s, 72 °C for 30 s and a final extension of 72 °C for 3 min.

The final product was quantified on the Qubit instrument using the Qubit Broad Range DNA kit (Invitrogen) and individual amplicons were pooled in equal concentrations. The pooled library was cleaned utilizing Ampure XP beads (Beckman Coulter) then the band of interest was further subjected to isolation via gel electrophoresis on a 1.5% Blue Pippin HT gel (Sage Science). The library was quantified via qPCR followed by 300-bp paired-end sequencing using an Illumina MiSeq instrument in the Genome Center DNA Technologies Core, University of California, Davis.

Each sample was processed using a custom workflow designed to produce the highest quality amplicon sequence variants (ASVs) for each sample. Analysis of raw sequence reads began by first barcode demultiplexing to sample and 16 S V3–V4 primer identification and trimming using dbcAmplicons (https://github.com/msettles/dbcAmplicons version 0.8.5). Any reads that could not be confidently assigned to sample by barcode, allowing one mismatch, or to primer sequences, allowing ≤4 mismatches using the Levenshtein distance as long as the final 4 bases of the primer perfectly matched the target sequence, were discarded. The resulting forward and reverse reads were preprocessed using HTStream (https://github.com/s4hts/HTStream version 1.2.0) to overlap reads into full V3–V4 sequences while also excluding any reads that contained a no-call ('N') character. Overlapped reads were then filtered, keeping amplicons ≥350 bp in

length, denoised, summarized to amplicon sequence variants (ASVs), and chimeric ASVs sequences removed using DADA2 (R version 3.6.1; DADA2 version 1.14.1)[70]. ASVs were then assigned to taxa using the RDP naive Bayesian classifier[71] implemented in DADA2 against the SILVA database reference of 16 S sequences[72,73]. ASVs were aligned using DECIPHER and a phylogenetic tree of ASVs was constructed using phangorn[74,75]. Finally, rarefaction curves were calculated using the rarecurve function from the R package vegan.

Relative abundance of phyla and classes were calculated by normalizing read counts to library size using the relative log expression (RLE) normalization factor[76] and then dividing normalized read counts in each phylum or class by the total number of normalized read counts in each sample to obtain a percentage of reads that belonged to each phyla and class. Analyses of differential abundance of individual ASVs was carried out using limma voom[47] in Bioconductor (www.bioconductor.org) modeling the effect of *Mecp2-e1* genotype longitudinally across disease course, stratified by sex and controlling for within-litter effects. This analysis generated overall longitudinal associations between genotype and individual ASVs, as well as associations between ASVs and genotype cross-sectionally at each time point. The Benjamini–Hochberg method was used to correct for multiple comparisons and obtain false discovery rates (FDRs). FDR values of less than 0.05 were considered statistically significant.

**Measurement of fecal cytokines**. Fecal pellets collected from *Mecp2-e1* mutant and wild-type littermate controls, at 6, 10, and 18 weeks for females and 6, 10, and 14 weeks for males, were analyzed. Fecal pellets were suspended in phosphate-buffered saline (PBS) with a protease inhibitor cocktail (PMSF #8553; Cell Signaling Technology, Danvers, MA), then physically disrupted by vortexing thoroughly followed by brief sonication. The homogenized pellets were centrifuged at 4 °C for 10 min to precipitate insoluble materials. Protein concentrations were measured using the BCA protein assay kit (Thermo Scientific, Waltham, MA). The quantification of cytokines was then determined using, the Milliplex™ MAP Mouse Cytokine/Chemokine Magnetic Bead Panel (Millipore, Burlington, MA). Samples were run in accordance with the manufacturer's protocol. Briefly, the samples were incubated with antibody-coupled beads then after a series of washes, a biotinylated detection antibody was added to the beads, and the reaction mixture was visualized by the addition of streptavidin conjugated to phycoerythrin. After the final washes the beads fluorescence was analyzed using a flow-based Luminex™ 100 suspension array system (Bio-Plex 200; Bio-Rad Laboratories, Hercules, CA). Each sample was analyzed in duplicate and results expressed as median fluorescent intensities (MFI). Finally, MFIs were converted to concentrations using the standard curve derived from the appropriate cytokines with the Bio Plex Manager software.

**Characterization of the fecal and cortical metabolome**. Fecal samples collected at 5, 9, and 19 weeks from females and 5, 9, and 16 weeks from males were processed and assessed for short chain fatty acids (SCFAs), biogenic amines and lipids, and cortical samples were processed and assessed for lipids at the West Coast Metabolomics Center (WCMC) at UC Davis. For SCFA analysis, fecal samples were processed by homogenization of 10 mg of fecal matter followed by solvent extraction using HPLC-grade ethylbutyric acid, hydrochloric acid, tert-butyl methyl ether (MTBE), and methylvaleric acid. SCFAs were measured using gas chromatography coupled with mass spectrometry (GCMS) on the Agilent GC7890B/5977MS with DB5 Duraguard 30 m × 0.25 mm × 0.25 u capillary column and measured against standards of known concentrations[77,78]. To analyze biogenic amines and lipids, samples were processed via homogenization followed by extraction with a bi-phasic solvent system comprised of methanol, MTBE, and water, which allowed for extraction of biogenic amines in the aqueous phase and lipids in the lipid phase[79]. To measure biogenic amines, polar phase lipid extraction by ultra high pressure liquid chromatography (UHPLC) was used coupled with mass spectrometer (MS). UHPLC was performed using an Agilent Infinity LC system with a Waters BEH Amide Column, and MS was carried out using a SCIEX Triple time of flight (TOF) 6600 mass spectrometer. To measure lipids, UHPLC was carried out on an Agilent 1290 Infinity LC system with a Waters CSH C18 column interfaced to a quadrupole TOF MS. Data were processed using an untargeted approach by first using mzMine 2.0 software to detect peaks and then collated with Agilent's MassHunter quantification method using retention time and exact mass, and the NIST14/Metlin/MassBank/Lipidblast libraries[80–82] to identify metabolites with manual confirmation of adduct ions and spectral scoring accuracy. To quantify data, peak heights were normalized using vector normalization to the sum of all peak heights for identified metabolites. Lipids underwent further quantification to obtain semi-quantitative data for known lipids. Quality control (QC) samples were included with each analysis.

Metaboanalyst (metaboanalyst.ca)[83] was used to further process data and carry out statistical analyses to compare fecal metabolites across genotype and stratified by sex, longitudinally across time, and to compare cortical lipids across genotype cross-sectionally at 19 weeks for females and 16 weeks for males. Prior to statistical analysis, data underwent filtering to remove metabolites that were higher than 25% relative standard deviation (standard deviation/mean) in QC samples and to remove metabolites that were nearly constant across the experiment and were within the interquantile range (IQR). Metabolites that passed filtering were then log transformed. For fecal metabolites, repeated measures two-way ANOVA was carried out to compare mutant vs. wild-type females and males longitudinally, and for cortical lipids, a t-test was carried out. FDRs were calculated using the

Benjamini–Hochberg method for each metabolite/lipid. An FDR < 0.05 was considered statistically significant for fecal metabolites, but a raw *p*-value threshold of 0.05 was used for cortical lipids. This was because we observed more subtle effects of *Mecp2-e1* genotype on lipids in the cortex than in fecal matter.

**Metabolomics enrichment analysis**. Metabolite enrichment analyses were carried out using ChemRICH[48], which utilizes chemical ontologies and chemical structures to cluster metabolites in a study-specific manner to yield enriched groups of metabolites. Biogenic amines and lipids that were annotated to specific compounds were then further annotated with InChIKeys, PubchemIDs, and SMILES using the Chemical Translation Service (http://cts.fiehnlab.ucdavis.edu/) and the PubChem Identifier Exchange Tool (https://pubchem.ncbi.nlm.nih.gov/idexchange/idexchange.cgi). Compounds that were able to be fully annotated and their genotype-associated FDR values and foldchange in mutant vs. wild-type mice were used for ChemRICH enrichment testing. Raw *p*-values were used in lieu of FDR values for cortical lipids since brain lipids showed more subtle differences. We considered clusters to be significantly enriched if FDR < 0.05 for fecal metabolomics and FDR < 0.10 for cortical lipids.

**Weighted gene co-expression network analysis (WGCNA) for fecal metabolomics**. To condense fecal metabolomics data into groups of co-regulated metabolites, we employed a Weighted Gene Co-Expression Network Analysis (WGCNA) to processed fecal metabolomics data. WGCNA analysis is commonly carried out to identify co-regulated gene networks using microarray and RNA-seq data[84], and here we applied it to metabolomics data to obtain modules of metabolite networks. Data were stratified by sex for the analysis. We selected a soft power threshold based on scale independence where the model fit ($R^2$) was 0.8, which was 18 for females and 5 for males (Supplementary Fig. 12). The minimum number of metabolites allowed in each network module was 10. The resulting network analysis resulted in 19 fecal metabolite modules for females and 33 for males (Supplementary Fig. 13). These resulting modules containing metabolites that were highly correlated with one another were then used in data integration to identify relationships between fecal metabolites, the fecal microbiome, and disease phenotypes. Hub metabolites were assigned using the WGCNA package and modules were designated based on their hub metabolite.

**Statistics and reproducibility**. All statistical analyses were carried out using R version 3.6.3 (www.r-project.org). Each mouse was considered a biological replicate and each analysis had between 6 and 12 mice/genotype/sex, depending on the measurement; details on exact N's are included in figure legends. To compare differences in phenotypic measurements between mutant and wild-type mice, we utilized linear mixed effects models for longitudinal data to control for repeated measures and with a random effect variable to control for within-litter effects. In addition to longitudinal analyses, we also carried out cross-sectional analyses at each time point and then used the Benjamini–Hochberg method to calculate FDRs and adjust for multiple comparisons across each time point. All analyses were stratified by sex in order to detect sex-specific effects of *Mecp2-e1* genotype.

To integrate microbiome data with phenotypic data, limma voom was used to test the effect of phenotype on ASVs, while controlling for genotype, age, and within-litter effects. To integrate metabolomics data with phenotypes, genotype, and age, linear mixed effects models were used to model the effect of each of these variables on the EigenValue for each metabolome module as calculated via WGCNA. These models contained random effects variables to correct for within-litter effects and repeated measures. Microbiome and metabolomics data was integrated by using limma voom to model the effect of metabolomic module EigenValues on ASVs while adjusting for genotype, age, and within-litter effects. To evaluate the relationship between fecal lipids and cortical lipids, Spearman's correlation coefficients were calculated. All analyses were corrected for multiple comparisons using the Benjamini–Hochberg method to calculate FDRs.

**Reporting summary**. Further information on research design is available in the Nature Research Reporting Summary linked to this article.

## Data availability
The numerical source data behind the graphs in the paper can be found in Supplementary Data 1-11. These data have also been deposited in Fighshare at https://doi.org/10.6084/m9.figshare.1690659786 Metabolomics data and 16 S sequencing data are also available on Figshare[85]. All other data are available within the main or supplementary files or can be obtained from the corresponding author on reasonable request.

## Code availability
Custom code used in the bioinformatic analysis of microbiome data is available at https://github.com/msettles/dbcAmplicons version 0.8.5. Custom code used in the statistical analysis of microbiome data and metabolomics data are available at https://github.com/karineier/Microbiome-Analyses and https://github.com/karineier/Metabolomics-Data-Integration.

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

## Acknowledgements

This work was funded by NIH R01 AA027075 to J.M.L. and the UC Davis Intellectual and Developmental Disabilities Research Center (IDDRC) [P50HD103526]. The sequencing was carried out by the DNA Technologies and Expression Analysis Cores at the UC Davis Genome Center and was supported by a NIH Shared Instrumentation Grant [1S10OD010786-01]. K.N. was supported by NICHD F32 HD105325.

## Author contributions

K.N. participated in study design, analyzed and interpreted the data, and drafted the manuscript. J.M.L. conceptualized and designed the study, obtained funding, aided in interpretation of data, and revised the manuscript. D.H.Y. conceptualized and designed the study, acquired data, assisted in data interpretation, and revised the manuscript. B.D.J., M.L.S., S.S.H., and P.A. participated in data analysis. T.E.G., R.L.P., D.C., S.M.H., K.M.Y., M.R., and A.M. acquired data.

## Competing interests

The authors declare no competing interests.
