## [Transparent Peer Review File · Communications Biology]

Reviewers' comments:

Reviewer #1 (Remarks to the Author):

The manuscript by Neier et al provides a comprehensive overview of the progression of microbiome alterations, metabolic changes and motor behavioral symptoms of males and females in an animal model of Rett syndrome. The manuscript is well-written, and the findings are interesting. A big strength of this paper is the characterization of RTT phenotypes in females. However, the conclusions of this paper should be toned-down. For example, in the abstract and in the discussion, the authors claim that their findings suggest new potential therapeutic targets for RTT. However, this study is descriptive and does not identify or tested new potential therapeutic targets for RTT. Furthermore, there are additional concerns outlined below that must be addressed before we can recommend publication in Communications Biology.

The fact that alterations in the gut microbiome and gut metabolome precede changes in behavior in this animal model is very intriguing. Could the authors add some more evidence including additional experiments characterizing gastrointestinal health (for example gastrointestinal transit and gastrointestinal permeability)? It would be interesting to see if gastrointestinal deficits precede the behavioral changes.

As mentioned, the characterization of RTT in females is a big differential in this study. Could the authors highlight that and expand their discussions on previous male- bias in RTT and microbiome literature?

The term "age-related" should be replaced by "time-dependent".

How the authors explain the peak of alterations in the gut microbiome in the weeks of 12 and 16 weeks in females and at 14 and 15 weeks in males?

The authors should fix their figures. There are several mistakes including random lines, data points and statistics bars, numbers and axis overlapping.

Figure 6 should mention analyte ids rather than colors.

Reviewer #2 (Remarks to the Author):

This is a descriptive study of male and female rodent (mouse) model of Rett Syndrome. The authors have profiled the gut microbiome and lipid metabolome of the mutated mice versus wild type mice from 5-6 weeks of age till 19 weeks of age. The authors have further performed correlational analysis of the symptoms compared to the lipid deficiencies of the brain and gut microbiome.

Rett Syndrome is a rare genetic disorder with no available cure. Understanding the disease-associated gut problems will be important for future studies.

Major concerns:

- The study has incremental impact for Rett Syndrome studies and is merely an observational research.
- The authors should show that correcting the problems associated with lipid metbolome can be recovered by targeting lipid metabolism by in vivo studies.
- The authors should also verify whether targeting lipid metabolism in male and female mutant mice can recover the identified gut microbial issues and brain lipide contents.

- Lovastatin has shown to improve the lifespan of the Mecp2 deficient male mice. The authors should show if lovastatin/similar treatment can expand the lifespan of this model of Rett Syndrome mice, and if any of the other neurological symptoms are rescued.

Reviewer #3 (Remarks to the Author):

Thank you for allowing me to review the manuscript by Neier and colleagues "Sex disparate gut microbiome and metabolome perturbations precede disease progression in a mouse model of Rett syndrome" an interesting study describing microbiome and metabolome studies and gender-related modification in Rett mouse model.

The study design is well-described, achieving interesting and novel conclusions that might be of interest for scientific community. My impression is positive although some clarifications might be warranted.

Fecal microbiome has been studied on weekly basis between 5 and 19 weeks of age in females mouse model vs wt. In legends of Table 2 is stated measurement done at, 5, 9, 19 weeks of age. Might you clarify please?

I would suggest to clarify the label in the first row of Table 2 and 3. I suggest to specify the altered metabolites instead of the key metabolite. How could you define the "key compound"? I see many unknown metabolites from supplementary table 13. Have you excluded these metabolites from your analysis? I suggest to include in the supplementary the metabolites increased and decreased along with t-test and FDR.

Have the mice (males and females) followed similar diet? Maybe the differences seen in females are not evidenced in males since they do not feed properly due to the severity of disease expression. MECP2 is also expressed in the gut and in nervous enteric system. The dysbiosis in females might be either due to a direct dysfunction of MECP2 in the gut or to the altered motility of the mice.

The inverse correlation between fecal and brain lipids concentrations is really interesting. Might you include also statistical significance along with correlation coefficient? The inverse correlation might include also the case of reduced level on lipid fecal metabolites vs high amount of the same metabolites in the brain. I suggest to specify/highlight the increased fecal metabolites and corresponding reduced one in brain. In figure 7B you showed the correlation between females at 9 weeks and brain at 19 weeks. Might you modify the figure title since might be misleading (brain lipids are analyzed at 19 weeks and not at 9 weeks). How could you explain that you do not have the highest degree of correlation at 19 weeks of age?

Minor

Please spell PC, PE, TAG, SM out in the different figures and tables.

Responses to Reviewers

We thank all reviewers for the positive comments and constructive feedback that we have addressed with additional experimental data and extensive revisions. Specifically, we now include new data characterizing gastrointestinal health longitudinally across disease progression my measurements of the levels of fecal cytokines (new Figure 4a-c, revised Fig 7a-b, and new Suppl Figure 7) and gut motility (new Suppl. Figure 8). Overall, there are two new main figures, two new Supplemental figures, and three new Supplemental tables in the revised manuscript. We have also added two new authors, Dr. Abdullah Madany and Dr. Paul Ashwood, because of their contributions to the cytokine analyses. The manuscript has been significantly improved by these and other edits suggested by the reviewers and summarized in the point-by-point rebuttal.

Reviewers' comments:

Reviewer #1 (Remarks to the Author):

The manuscript by Neier et al provides a comprehensive overview of the progression of microbiome alterations, metabolic changes and motor behavioral symptoms of males and females in an animal model of Rett syndrome. The manuscript is well-written, and the findings are interesting. A big strength of this paper is the characterization of RTT phenotypes in females. However, the conclusions of this paper should be toned-down. For example, in the abstract and in the discussion, the authors claim that their findings suggest new potential therapeutic targets for RTT. However, this study is descriptive and does not identify or tested new potential therapeutic targets for RTT. Furthermore, there are additional concerns outlined below that must be addressed before we can recommend publication in Communications Biology.

Thank you for taking the time to carefully review our manuscript and for the constructive feedback. As suggested, we have toned down the conclusions in the Abstract and in the Discussion and refocused our conclusions to highlight the importance of our female-specific findings. Altered text is highlighted in yellow.

Abstract

P. 2, L. 35-37: “These findings identify new molecular pathways of RTT disease progression and demonstrate the importance of further study in female animal models.”

Discussion

P. 28, L. 730-743: “The findings from this study provide insights into the underlying molecular pathways involved in RTT disease progression and demonstrate the sexually dimorphic nature of RTT pathophysiology. We found a convergence of metabolism, gut microbial communities and cytokine profiles, and neuromotor phenotypes in a patient-relevant mouse model of RTT. In females, differences in metabolites and microbes between *Mecp2-e1* mutant and wild-type were observed prior to onset of neuromotor phenotypes, suggesting that these pathways play a role in disease pathology. Critically, our data demonstrated that female *Mecp2-e1* mutant mice recapitulate molecular signatures of human RTT more consistently than male *Mecp2-e1* mutants, including alterations in the gut microbiome and cytokine profiles. We also discovered that lipid malabsorption may contribute to RTT neuromotor phenotypes in females, a potential new pathway that should be studied further for potential therapies. Future animal studies

investigating RTT disease pathology and treatments should include females in order to effectively translate findings to human RTT patients.”

The fact that alterations in the gut microbiome and gut metabolome precede changes in behavior in this animal model is very intriguing. Could the authors add some more evidence including additional experiments characterizing gastrointestinal health (for example gastrointestinal transit and gastrointestinal permeability)? It would be interesting to see if gastrointestinal deficits precede the behavioral changes.

We agree that additional data on gut health would strengthen the paper. To address this, we measured cytokine levels in fecal samples across disease course at 6, 10, 14 (males), and 18 weeks (females), and included data on fecal pellet passage rate across disease course (collected weekly). Fecal cytokine levels are indicative of gut inflammation, and fecal pellet passage is indicative of gut motility. These data were collected from the same animals included in the rest of the study, allowing us to accurately assess timing of differences in this data relative to other observed phenotypes and molecular signatures. Below is the text included for this new data:

Results

P. 13-14, L. 308-354:

Gut inflammation and motility are impacted in Mecp2-e1 mutants in a sex-specific manner

An altered gut microbiome may be associated with gut inflammation and impaired motility in Mecp2-e1 mutant mice. Thus, we assessed longitudinal gut inflammation by measuring cytokine levels in fecal samples from wild-type and mutant Mecp2-e1 mice at 6, 10, and either 14 (males) or 18 weeks of age (females). We found that female and male mutant mice had distinctly altered fecal cytokine profile across disease progression. While decreased fecal interferon-gamma (IFN γ) levels were observed in both male and female mutant mice, interleukin-4 (IL-4) was higher in female mutants but lower in male mutants relative to wild-type (Figure 4a-b; Supplemental Table 12). Furthermore, genotype-related changes in IL-4 and IFN γ were present throughout disease progression beginning as early as 6 weeks in males. In mutant females, lower IFN γ levels were not observed until 10 weeks and higher IL-4 levels were only observed in later stages of disease progression at 18 weeks. Since IFN γ is mainly associated with T helper 1 (Th1) cell and IL-4 associated with T helper 2 (Th2) cell immune responses, these results indicate that female Mecp2-e1 mutants exhibited a shift towards a Th2 response at 18 weeks of age. In addition, longitudinally, female mutants exhibited increased fecal IL-1a, a pro-inflammatory cytokine, compared to wild-type controls, however there were no differences between mutant and wild-type IL-1a levels in males (Figure 4c). Mecp2-e1 mutant males also had decreased fecal IL-17 levels across disease course compared to wild-type littermates (Supplemental Figure 7, Supplemental Table 12). In females, IL-17 was mostly absent with few samples with detectable IL-17 in females. Mecp2-e1 mutant females also had decreased keratinocytes-derived chemokine (KC) levels compared to wild-type littermates at 10 weeks of age and increased relative KC levels at 18 weeks of age. In males there were few samples with detectable levels of KC (Supplemental Figure 7, Supplemental Table 12).

In addition to gut inflammatory cytokines, we also assessed gut motility by measuring the number of fecal pellets passed by each mouse within a five-minute period. Both female and male mutants passed more fecal pellets than wild-type controls longitudinally, though males did not have any significant differences between mutant and wild-type cross-sectionally

(Supplemental Figure 8). Female *Mecp2-e1* mutants displayed increased fecal pellet passage as early as 6 weeks of age, but there was not a consistent difference between mutant and wild-type littermates at all time points (Supplemental Figure 8).

Discussion

P. 25, L. 651-667:

We found that *Mecp2-e1* mutant females, but not males, exhibited a Th2-type response detectable in the fecal pellet in late stages of disease progression, as demonstrated by increased levels of IL-4 and decreased levels of IFN γ relative to wild-type controls. This is consistent with a clinical study that found increased IL-4 and decreased IFN γ in plasma from RTT patients compared to healthy controls⁵⁹. Interestingly, taken together these data suggest that a Th2 response is a species-independent feature of female MeCP2 deficiency and RTT disease progression. Persistent Th2 responses in the gut have been shown to result in chronic inflammation in other disease processes such as ulcerative colitis⁶⁰. The relative increase in fecal IL-4 levels observed in *Mecp2-e1* mutant females did not emerge until later in RTT disease progression, indicating that gut inflammation may be secondary or in response to alterations in the gut microbiome or metabolism. Further assessment of immune responses in the mucosal layer and at intestinal lymph nodes will be undertaken in follow up studies. In contrast, *Mecp2-e1* mutant males showed relatively decreased levels of fecal IFN γ , IL-4, and IL-17 throughout disease progression. This could indicate a dampened gut immune responsiveness by the complete loss of MeCP2e1 in males that is already present in early stages of disease.

As mentioned, the characterization of RTT in females is a big differential in this study. Could the authors highlight that and expand their discussions on previous male-bias in RTT and microbiome literature?

We appreciate that the reviewer pointed out the importance of the differential findings in males and females in this study, and we have highlighted the importance of these findings within the context of previous RTT studies in the manuscript in the Discussion. Altered text is highlighted in yellow.

P. 24, L. 627-632: “Cumulatively, our findings experimentally confirmed the relationship between RTT and the gut microbiome from human RTT patients and further demonstrate that *Mecp2* deficient female mice are a better model of human RTT patients than male mice. If this study had only included males, our findings would not have been consistent with microbiome, metabolite, and immune findings from RTT patients.”

P. 27, L. 711-714: “Most previous studies in animal models of RTT have focused on males hemizygous for mutated or null *Mecp2*, and while critically informative for understanding the function of MeCP2, this resulted in gaps in understanding the complexity of RTT disease progression.”

P. 28, L. 730-732: “The findings from this study provide insights into the underlying molecular pathways involved in RTT disease progression and demonstrate the sexually dimorphic nature of RTT pathophysiology.”

P. 28, L. 736-739: “Critically, our data demonstrated that female *Mecp2-e1* mutant mice recapitulate molecular signatures of human RTT more consistently than male *Mecp2-e1* mutants, including alterations in the gut microbiome, metabolome, and cytokine profiles.”

P. 28, L. 742-743: “Future animal studies investigating RTT disease pathology and treatments should include females in order to effectively translate findings to human RTT patients.”

The term “age-related” should be replaced by “time-dependent”.

We have replaced “age-related” with “time-dependent” in the text of the manuscript.

How the authors explain the peak of alterations in the gut microbiome in the weeks of 12 and 16 weeks in females and at 14 and 15 weeks in males?

Thank you for pointing out the importance of this observation. We have included the following text in the discussion (P. 24-25, L. 635-650) to address this:

“Although *Mecp2-e1* mutant females and males exhibited altered gut microbes beginning early in disease progression at 5 weeks of age, the largest numbers of differentially abundant taxa were observed in later stages of disease. In females, the number of differentially abundant taxa fluctuated by over 100 taxa throughout disease course, indicating that the gut microbiome undergoes dynamic changes throughout RTT disease progression in females. The number of differentially abundant taxa peaked at 12 (188 ASVs) and 16 weeks (184 ASVs) in females. The peak differences at 12 weeks may have been a response in the gut microbiome to onset of neurological and gait abnormalities (at 10 and 11 weeks, respectively), and the peak at 16 weeks coincides with the observed fecal cytokine response, indicative of gut inflammation. In males, on the other hand, there was a gradual, steady increase in the number of differentially abundant taxa with disease progression. The peak occurred at 14 (86 ASVs) and 15 weeks (99 ASVs), but there was a similarly large number of differentially abundant taxa at 16 weeks as well (76 ASVs). Additional studies are needed to elucidate what precisely contributes to the timing of peak differences in gut microbe abundance in RTT.”

The authors should fix their figures. There are several mistakes including random lines, data points and statistics bars, numbers and axis overlapping.

Thank you for noticing this – we believe this was a formatting error that has been fixed in the revised submission.

Figure 6 should mention analyte ids rather than colors.

We have updated Figure 6, the legend for Figure 6, and the manuscript text describing Figure 6 to designate modules by hub metabolites, as identified using WGNCA, rather than colors.

Reviewer #2 (Remarks to the Author):

This is a descriptive study of male and female rodent (mouse) model of Rett Syndrome. The authors have profiled the gut microbiome and lipid metabolome of the mutated mice versus wild type mice from 5-6 weeks of age till 19 weeks of age. The authors have further performed correlational analysis of the symptoms compared to the lipid deficiencies of the brain and gut microbiome.

Rett Syndrome is a rare genetic disorder with no available cure. Understanding the disease-associated gut problems will be important for future studies.

Major concerns:

- The study has incremental impact for Rett Syndrome studies and is merely an observational research.

Thank you for taking the time to review our manuscript. This research is indeed largely descriptive aside from using a mouse model with mutated *Mecp2-e1*. However, we respectfully disagree that the impact for Rett syndrome is incremental. This was the first study investigating the gut microbiome and metabolome longitudinal changes during disease progression in RTT, which has led to multiple important new discoveries and insights with translational relevance. This study addressed several critical gaps in the RTT literature as indicated within the Introduction, including but not limited to use of a patient-relevant mouse model, and measurement in both male and female mice. To better summarize the strengths of this study, we have included the following sentence in the revised Discussion.

P. 21, L. 548-553: “The innovative nature of this study exploits longitudinal measurements in the same mice over disease course which allowed us to 1) identify molecular pathways that emerge prior to overt neuromotor phenotypes, 2) integrate multi-omic data to show identify connections between neuromotor and metabolic pathways in RTT, and 3) utilized bioavailable tissues (i.e., fecal samples) that can be obtained in human studies, allowing for direct comparisons between this study and those carried out in RTT patients.”

Because we have not specifically tested therapies, as also pointed out by Reviewer 1, we agree that our original conclusions were too strong and have thus refocused them accordingly in the revised Discussion.

P. 28, L. 730-743: “The findings from this study provide insights into the underlying molecular pathways involved in RTT disease progression and demonstrate the sexually dimorphic nature of RTT pathophysiology. We found a convergence of metabolism, gut microbial communities and cytokine profiles, and neuromotor phenotypes in a patient-relevant mouse model of RTT. In females, differences in metabolites and microbes between *Mecp2-e1* mutant and wild-type were observed prior to onset of neuromotor phenotypes, suggesting that these pathways play a role in disease pathology. Critically, our data demonstrated that female *Mecp2-e1* mutant mice recapitulate molecular signatures of human RTT more consistently than male *Mecp2-e1* mutants, including alterations in the gut microbiome and cytokine profiles. We also discovered that lipid malabsorption may contribute to RTT neuromotor phenotypes in females, a potential new pathway that should be studied further for potential therapies. Future animal studies investigating RTT disease pathology and treatments should include females in order to effectively translate findings to human RTT patients.”

- The authors should show that correcting the problems associated with lipid metabolome can be recovered by targeting lipid metabolism by in vivo studies.

See cumulative response below

- The authors should also verify whether targeting lipid metabolism in male and female mutant mice can recover the identified gut microbial issues and brain lipide contents.

See cumulative response below

- Lovastatin has shown to improve the lifespan of the *Mecp2* deficient male mice. The authors should show if lovastatin/similar treatment can expand the lifespan of this model of Rett Syndrome mice, and if any of the other neurological symptoms are rescued.

We agree with the reviewer that these additional studies mentioned in this comment and the two preceding it would be valuable in both confirming the role for lipid metabolism in RTT disease pathology and in identifying potential new therapies for RTT. However, we are not convinced that Lovastatin is the best choice to treat the lipid metabolic changes, because our data suggest that the increased levels of lipids in feces are due to malabsorption since they are associated with lower lipid levels in brain. We do plan to carry out treatment studies in future work, but they are outside the scope of the current study. Within the present study, we aimed to provide a comprehensive longitudinal characterization of RTT disease progression with a highlight on the gut microbiome and metabolome in both male and female mice. The findings from this study will serve as a useful resource for other human and animal RTT studies, and has elucidated new molecular pathways involved in RTT disease progression.

Reviewer #3 (Remarks to the Author):

Thank you for allowing me to review the manuscript by Neier and colleagues “Sex disparate gut microbiome and metabolome perturbations precede disease progression in a mouse model of Rett syndrome” an interesting study describing microbiome and metabolome studies and gender-related modification in Rett mouse model.

The study design is well-described, achieving interesting and novel conclusions that might be of interest for scientific community. My impression is positive although some clarifications might be warranted.

Fecal microbiome has been studied on weekly basis between 5 and 19 weeks of age in females mouse model vs wt. In the legends of Table 2 is stated measurement done at, 5, 9, 19 weeks of age. Might you clarify please?

Table 2 presents the results of a chemical enrichment analysis based on fecal metabolomic data, which was carried out at 5, 9, and 19 weeks of age. We clarified this in the text of the manuscript:

P. 14-15, L. 357-372: “Since microbial communities in the gut produce a variety of key metabolites, such as short-chain fatty acids (SCFAs) relevant to nervous system metabolism, we tested the hypothesis that fecal metabolites may also precede disease phenotypes through longitudinal comparisons of *Mecp2-e1* mutant and wild-type control littermates by measuring fecal metabolites at 5, 9, 16 (males), and 19 (females) weeks of age.”

I would suggest to clarify the label in the first row of Table 2 and 3. I suggest to specify the altered metabolites instead of the key metabolite. How could you define the “key compound”? I see many unknown metabolites from supplementary table 13. Have you excluded these metabolites from your analysis? I suggest to include in the supplementary the metabolites increased and decreased along with t-test and FDR.

We specified the altered metabolites within Supplemental Tables 13 and 17, as listing each compound within Tables 2 and 3 would result in an extremely large and difficult to read table. The p-values and foldchange values are also included in these new Supplemental Tables. We also explained the definition of a “key compound” within the tables – the compound with the lowest p-value – within the legends of Tables 2 and 3.

Unknown compounds were excluded from the ChemRICH analysis. This is explained in the following text on P. 15-16, L. 392-405:

“Biogenic amines and lipids that could be accurately identified as known compounds and annotated to metabolite databases (89 and 169, respectively) were used in enrichment testing using ChemRICH⁴⁸ to identify clusters of related metabolites that were longitudinally associated with *Mecp2-e1* genotype.”

Have the mice (males and females) followed similar diet? Maybe the differences seen in females are not evidenced in males since they do not feed properly due to the severity of disease expression. MECP2 is also expressed in the gut and in nervous enteric system. The dysbiosis in females might be either due to a direct dysfunction of MECP2 in the gut or to the altered motility of the mice.

While all mice were fed the same diet, gut microbiome and metabolism differences between males and females could potentially be due to food intake or energy expenditure differences. We have previously found that *Mecp2-e1* mutant males exhibited decreased food intake and energy expenditure compared to wild-type males, whereas mutant and wild-type females exhibited nonsignificant changes in food intake and energy expenditure. We included some text explaining this in the Discussion.

P. 27-28, L. 721-727: “Male, but not female, *Mecp2-e1* mice exhibited significantly reduced food intake and energy expenditure in previous studies³⁸, which could also contribute to the observed sex-specific effects on the gut microbiome and metabolism.”

The inverse correlation between fecal and brain lipids concentrations is really interesting. Might you include also statistical significance along with correlation coefficient? The inverse correlation might include also the case of reduced level on lipid fecal metabolites vs high amount of the same metabolites in the brain. I suggest to specify/highlight the increased fecal metabolites and corresponding reduced one in brain.

Statistical significance is represented in Figure 7B, with carrots (^) denoting relationships statistically significant at FDR<0.10. We included this in the figure legend.

Since dietary lipids undergo processing in the gut and liver before being transported to the brain, we did not observe strong associations between the same lipid in fecal matter and in the brain. Instead, we observed strong relationships between fatty acids, two in particular, and multiple critical brain lipids including sphingomyelins, phosphatidylcholines, and phosphatidylethanolamines. We included a statement describing this in the Results section:

P. 20, L. 532-537: “Many of the lipids that were decreased in *Mecp2-e1*^{-/-} cortex relative to *Mecp2-e1*^{+/+} female littermates are critical for neuronal function, such as phosphatidylethanolamines, sphingomyelins, and phosphatidylcholines^{51–53}. Relative decreases in the cortical levels of these important lipids showed especially strong correlations with relative increases in fecal levels of fatty acid 18:4 and 18:2 (**Figure 8b**).”

In figure 7B you showed the correlation between females at 9 weeks and brain at 19 weeks. Might you modify the figure title since might be misleading (brain lipids are analyzed at 19 weeks and not at 9 weeks). How could you explain that you do not have the highest degree of correlation at 19 weeks of age?

Thank you for suggesting this clarification. We have updated the Figure title to “Correlation Between Fecal Lipids at 9 weeks and Brain Lipids at 19 weeks.”

To address the strong association between fecal lipids at 9 weeks and brain lipids at 19 weeks, we added the following text to the Discussion:

P. 26-27, L. 699-703: “Notably, the strongest correlations between cortical (19 weeks) and fecal lipid levels were with fecal lipid levels at 9 weeks. Decreased fecal lipid levels in *Mecp2-e1* mutant females were the most pronounced at 9 weeks, but may have resulted in long-term impacts on brain lipid levels.”

Minor

Please spell PC, PE, TAG, SM out in the different figures and tables.

Spelling out these acronyms within the figures results in a large amount of text on axes and the text then needs to be adjusted to be much smaller and becomes difficult to read. Thus, we spelled out each acronym within the figure legends and tables.

REVIEWERS' COMMENTS:

Reviewer #1 (Remarks to the Author):

The authors have addressed all of my concerns with the original manuscript